# Directional Neural Collapse Explains Few-Shot Transfer in Self-Supervised Learning

**Achleshwar Luthra** [* 1]   **Yash Salunkhe** [* 1]   **Tomer Galanti** [* 1]

## Abstract

Frozen self-supervised representations often transfer well with only a few labels across many semantic tasks. We argue that a single geometric quantity, *directional* CDNV (decision-axis variance), sits at the core of two favorable behaviors: strong few-shot transfer within a task, and low interference across many tasks. We show that both emerge when variability *along* class-separating directions is small. First, we prove sharp non-asymptotic multiclass generalization bounds for downstream classification whose leading term is the directional CDNV. The bounds include finite-shot corrections that cleanly separate intrinsic decision-axis variability from centroid-estimation error. Second, we link decision-axis collapse to multitask geometry: for independent balanced labelings, small directional CDNV across tasks forces the corresponding decision axes to be nearly orthogonal, helping a single representation support many tasks with minimal interference. Empirically, across SSL objectives, directional CDNV collapses during pretraining even when classical CDNV remains large, and our bounds closely track few-shot error at practical shot sizes. Additionally, on synthetic multitask data, we verify that SSL learns representations whose induced decision axes are nearly orthogonal. The code and project page of the paper are open-sourced.

## 1. Introduction

Self-supervised learning (SSL) (Balestriero et al., 2023) has become a standard way to pretrain visual and multimodal representations without labels . A striking empirical fact is that frozen SSL features often enable strong few-shot

---
[*]Equal contribution  [1]Department of Computer Science & Engineering, Texas A&M University. Correspondence to: Tomer Galanti <galanti@tamu.edu>.

*Proceedings of the $43^{rd}$ International Conference on Machine Learning*, Seoul, South Korea. PMLR 306, 2026. Copyright 2026 by the author(s).

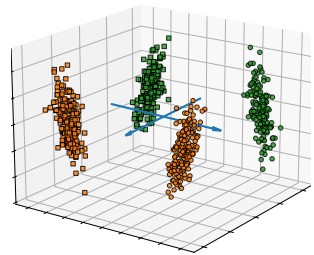

*Figure 1.* **Directional collapse and multitask orthogonalization in SSL.** Self-supervised pretraining suppresses within-class variance *along* class-separating directions (small directional CDNV) while leaving substantial variance in orthogonal, task-irrelevant subspaces. When directional CDNV is small for multiple independent labelings, the induced decision axes become nearly orthogonal, enabling a single representation to support many tasks with low interference.

transfer, with only a handful of labeled examples per class, across multiple downstream tasks simultaneously. Yet we still lack a clean geometric explanation for when and why this behavior occurs. This paper asks:

> ***Given a fixed SSL representation, what geometric properties enable effective few-shot adaptation across multiple tasks simultaneously?***

A useful point of comparison is supervised training, where representation geometry is better understood. In many supervised classifiers, late-layer features exhibit *neural collapse* (NC) (Papyan et al., 2020; Han et al., 2022): within each class, embeddings concentrate near a single mean; across classes, means spread in an approximately simplex-like configuration; and the linear classifier aligns with these directions. This geometry directly *relates* to few-shot success: when within-class dispersion is small relative to between-class separation, simple downstream rules such as nearest-class-centroid (NCC) achieve low error from limited data. Empirically, few-shot transfer and downstream transferability correlate with reduced within-class variability relative to between-class separation, as observed in analyses of few-shot representations and neural-collapse-inspired transferability metrics on the target data (Goldblum et al., 2020;

Galanti et al., 2022b; Wang et al., 2023; Li et al., 2024).

Empirically, SSL representations do exhibit clustering and separability with respect to semantic labels (Chen et al., 2020; Caron et al., 2021; Oquab et al., 2024; Ben-Shaul et al., 2023; Weng et al., 2025), even though labels are never used during pretraining. For example, (Ben-Shaul et al., 2023) shows that SSL training induces semantic label-based clustering and increasingly strong nearest-class-centroid (NCC) separability, this effect strengthens during training and in deeper layers and appears simultaneously for multiple labelings in parallel. These findings suggest an NC-like phenomenon in SSL, but also highlight a key difference: the geometry that matters for decisions can improve without requiring global within-class collapse in all directions.

SSL, however, is not trained to induce global within-class collapse. Because labels are absent during pretraining, there is no direct pressure to reduce *total* within-class variance. Empirically, SSL embeddings are often strongly *anisotropic*: substantial variance persists in directions that do not affect class decisions (e.g., nuisance or augmentation-induced directions), while the directions that *do* separate classes can be well organized. As a result, global clustering proxies such as the class-distance-normalized-variance (CDNV) (Galanti et al., 2022b) can be too coarse and yield pessimistic or misleading predictions of few-shot transfer.

Recent work begins to address this mismatch by measuring variability only *along class-separating directions* (Luthra et al., 2025b). They introduce a *directional* analogue of CDNV that keeps the component of within-class variance that perturbs the decision margin while ignoring variance in orthogonal subspaces. This is the right geometric target for anisotropic SSL representations, and it suggests that few-shot error can be small even when total variance remains large. However, existing bounds based on this quantity are often loose (and sometimes vacuous) at practical shot sizes, and the empirical evidence is limited to a narrow set of settings (e.g., ResNet-50 with SimCLR).

## 1.1. Contributions

We study few-shot transfer from a fixed self-supervised representation through variability *along* class-separating directions. Our main contributions are:

- **Sharp few-shot guarantees driven by decision-axis variability.** We prove non-asymptotic multiclass error bounds for nearest-class-centroid (NCC) and linear probing (LP) classification whose leading term is governed by *directional* CDNV (decision-axis variance), rather than classical CDNV which aggregates variance over all directions (as in (Galanti et al., 2022b;a; 2023b)). Our analysis makes the shot dependence explicit through finite-sample centroid-estimation terms and a fourth-moment

correction for heavy tails, yielding realistic error estimates across a wide range of shot sizes (Theorems C.2 and 4.1).

- **Decision-axis collapse yields accurate, non-vacuous few-shot certificates.** Across diverse SSL encoders, and objectives, we find that decision-axis variability collapses strongly during SSL training even when total within-class variance remains large, revealing a pervasive *anisotropic* geometry where most variability lies in directions that do not affect discrimination. In this regime, our bound closely tracks observed few-shot error and is substantially more informative than clustering-based proxies and prior directional bounds that can be loose at practical shot sizes (e.g., (Luthra et al., 2025b)). On the theory side, we show that the leading coefficient in front of $\tilde{V}_{ij}$ is *optimal*: in the known-centroid limit, pairwise NCC error is a one-dimensional tail event along the separating axis, and Cantelli's inequality (with its tight two-point extremizer) implies that no distribution-free second-moment bound can improve the factor $4$ (App. D).

- **Multitasks geometry: small decision-axis variance forces near-orthogonality of decision directions.** We prove a structural consequence of simultaneously small decision-axis variability across tasks: for two independent balanced binary labelings, small directional CDNV forces the corresponding decision axes to be nearly orthogonal (Prop. 4.2). We complement this with a simple factor model showing how a single representation can support many tasks with small decision-axis variance even when classical CDNV is large, since most within-class energy can concentrate in directions orthogonal to all task-relevant axes (Sec. 4.3). Empirically, on controlled synthetic data with independent visual factors (e.g., shape, size, color, pattern), we find that SSL encoders map distinct factors to approximately orthogonal directions, consistent with this multitask prediction.

## 2. Related Work

**Clustering properties in SSL.** Although SSL is trained without labels, learned representations often align with semantic categories and support simple downstream rules. Empirically, this occurs even in objectives that do not explicitly cluster representations: contrastive and non-contrastive pretraining can yield features with strong $k$-NN and linear-probe performance and increasingly clear semantic structure in deeper layers and later training (Chen et al., 2020; Caron et al., 2021; Ben-Shaul et al., 2023; Weng et al., 2025; Oquab et al., 2024). These findings motivate geometric explanations of transfer that characterize how class-conditional distributions are arranged in embedding space, rather than focusing only on instance discrimination.

At the same time, SSL representations are often strongly *anisotropic*: substantial within-class variability can persist in nuisance or augmentation-induced directions that have little effect on discrimination. This behavior has been documented empirically and theoretically in analyses of contrastive and non-contrastive objectives and alignment–uniformity trade-offs (Wang & Isola, 2020; Wang & Liu, 2021; Chen et al., 2021; HaoChen et al., 2021; Arora et al., 2019), and is also implicit in the design of modern SSL methods that explicitly discourage global collapse by preserving variance and decorrelating features (Zbontar et al., 2021; Bardes et al., 2022). As a result, global clustering proxies that aggregate variance across all directions can be too coarse to reliably predict few-shot performance.

**Theoretical analyses of SSL.** A growing literature studies why self-supervised learning works. Early analyses of contrastive learning related InfoNCE-like objectives to mutual information (Bachman et al., 2019), while later work noted that strict MI constraints can be overly restrictive (McAllester & Stratos, 2020; Tschannen et al., 2020). Another influential line characterizes representations via *alignment* and *uniformity* (Wang & Isola, 2020; Wang & Liu, 2021; Chen et al., 2021). These descriptors are informative, but they do not directly specify how *different* semantic classes are arranged.

This gap motivated analyses of when contrastive objectives recover clusters or latent structure (Arora et al., 2019; Tosh et al., 2021; Zimmermann et al., 2021; Ash et al., 2022; Nozawa & Sato, 2021; HaoChen et al., 2021; 2022; Shen et al., 2022; Wang et al., 2022; Awasthi et al., 2022; Bao et al., 2022), often under strong assumptions such as conditional independence of augmentations given a latent cluster (Arora et al., 2019; Saunshi et al., 2022; Tosh et al., 2021; Awasthi et al., 2022). Recent work relaxes aspects of this picture via alternative function-class biases (HaoChen & Ma, 2023) or capacity limitations that discourage splitting semantic clusters under InfoNCE (Parulekar et al., 2023). A complementary perspective connects SSL objectives to supervised criteria in simplified settings (Balestriero & LeCun, 2024; Luthra et al., 2025b;a).

Our work is closest to analyses that certify *downstream* performance from frozen representations. Most closely, Luthra et al. (2025b) introduce *directional* CDNV, which measures within-class variability only along class-separating axes and yields an anisotropy-aware certificate for NCC. We build on this directional viewpoint in three ways: (i) we give sharper, non-asymptotic multiclass guarantees for both NCC and LP whose leading term is governed by directional CDNV, including explicit finite-shot centroid-estimation corrections and a separate fourth-moment term isolating tail effects, with an optimal leading constant under second-moment information; (ii) we provide broad empirical evi-dence that decision-axis variability collapses across diverse SSL paradigms (contrastive, non-contrastive, masked prediction, distillation, and multimodal pretraining), suggesting that suppressing variance along discriminative directions is an implicit, method-agnostic outcome even when total within-class variance remains large; and (iii) we connect this anisotropic geometry to multitask structure by showing that if directional CDNV is simultaneously small for independent balanced labelings, then the associated decision directions must be nearly orthogonal, formalizing how a single representation can support many tasks while classical CDNV stays large.

Beyond these strands, theory has examined learning dynamics in simplified models (Tian, 2022; Ji et al., 2023; Wen & Li, 2021; Tian, 2023), the role of augmentations (Tian et al., 2020; Feigin et al., 2025), projection heads (Gupta et al., 2022; Gui et al., 2023; Xue et al., 2024; Ouyang et al., 2025), sample complexity (Alon et al., 2024; Yuan et al., 2022), and links between contrastive and non-contrastive methods (Wei et al., 2021; Balestriero & LeCun, 2022; Lee et al., 2021; Garrido et al., 2023; Shwartz-Ziv et al., 2023).

**Neural collapse and geometric predictors of transfer.** Neural collapse (NC) (Papyan et al., 2020; Han et al., 2022) describes a common end-of-training geometry in supervised classifiers: (i) *within-class collapse*, where embeddings concentrate near class means; (ii) *class separation*, where means form a nearly simplex equiangular tight frame (ETF); and (iii) *alignment* between classifier weights and feature geometry. NC has motivated geometric predictors of generalization and transfer, and neural-collapse-inspired measures on the *target* dataset have been shown to correlate with transferability across pretrained models (Wang et al., 2023). In few-shot regimes, representation analyses likewise emphasize the role of within-class compactness relative to between-class separation for centroid-based decision rules (Goldblum et al., 2020; Galanti et al., 2022b). Our goal is to extend this NC perspective to SSL in a way that respects anisotropy: rather than requiring global collapse, we characterize an SSL analogue that demands "collapse" only along decision-relevant directions, and we derive practical few-shot certificates that match this geometry.

## 3. Problem Setup

The primary goal of self-supervised learning (SSL) is to learn representations that are easily adaptable to downstream tasks. We formalize the conditions under which an SSL-trained representation $f : \mathcal{X} \to \mathbb{R}^d$ supports accurate few-shot classification. We say that $f$ is *good* if, after seeing only a few labeled examples per class, it yields low test error on the downstream task. We consider training an embedding function $f \in \mathcal{F} \subset \{ f \mid f : \mathcal{X} \to \mathbb{R}^d \}$ via self-supervised learning, using an unlabeled dataset $X =$

$\bigcup_{c=1}^{C} \{x_{c,i}\}_{i=1}^{n} \subset \mathcal{X}$ (e.g., images). Our goal is to learn a *"meaningful"* function $f$ that maps samples to $d$-dimensional embedding vectors.

For instance, we can call a representation meaningful if the hidden class labels are recoverable from the samples. Let $f : \mathcal{X} \to \mathbb{R}^d$ be a fixed representation learned by SSL. Fix a subset of classes $\mathcal{C} \subseteq [C]$ with $|\mathcal{C}| = C'$. For evaluation, draw $c \sim \mathrm{Unif}(\mathcal{C})$, then draw $x \sim D_c$, and set the label $y := c$. Let $D_{\mathcal{C}} := \frac{1}{C'} \sum_{c \in \mathcal{C}} D_c$ denote the balanced mixture over the selected classes. Given a classifier $h : \mathbb{R}^d \to \mathcal{C}$, its error on this mixture is

$$\mathrm{err}_{D_{\mathcal{C}}}(h) := \Pr_{c \sim \mathrm{Unif}(\mathcal{C}), \, x \sim D_c} \big( h(f(x)) \neq c \big).$$

For each $c \in \mathcal{C}$, draw an i.i.d. support set $\widehat{S}_c := \{(x_{c,s}, c)\}_{s=1}^{m} \sim D_c^m$, independent across classes and independent of the test draw $(c, x)$. Let $\widehat{S} := \bigcup_{c \in \mathcal{C}} \widehat{S}_c$. A downstream classifier $h_{f,\widehat{S}}$ is trained on $\widehat{S}$ and uses the features $f(x)$ at test time.[1] The expected $m$-shot error of $f$ is $\mathrm{err}_{m,\mathcal{C}}(f) := \mathbb{E}_{\widehat{S}_1, \dots, \widehat{S}_{C'}} [\mathrm{err}_{D_{\mathcal{C}}}(h_{f,\widehat{S}})]$.

**Downstream classifiers.** We study two simple downstream classifiers on top of the frozen encoder $f$: a linear probe (LP) and the nearest class centroid (NCC) rule. Given a support set $\widehat{S} = \{(x_{c,s}, c) : c \in \mathcal{C}, \ s \in [m]\}$ and a test feature $z = f(x) \in \mathbb{R}^d$, define

$$\widehat{y}^{\triangle}(z) \in \mathcal{C} \quad \text{and} \quad \mathrm{err}_{m,\mathcal{C}}^{\triangle}(f) := \frac{1}{|\mathcal{C}|} \sum_{i \in \mathcal{C}} \Pr \big( \widehat{y}^{\triangle}(z_i) \neq i \big),$$

where $\triangle \in \{\mathrm{LP}, \mathrm{NCC}\}$ and $z_i := f(x_i)$ with $x_i \sim D_i$ independent of $\widehat{S}$.

The LP is the minimizer of the population 0–1 error over all linear rules $g(z) = \arg\max_{c \in \mathcal{C}} (w_c^\top z + b_c)$ on top of $f$, estimated from the support set $\widehat{S}$. For NCC, let $z_{c,s} := f(x_{c,s})$, $\mu_c := \mathbb{E}_{x \sim D_c}[f(x)]$, and $\widehat{\mu}_c := \frac{1}{m} \sum_{s=1}^{m} z_{c,s}$ with $\delta_c := \widehat{\mu}_c - \mu_c$. Given a test feature $z = f(x) \in \mathbb{R}^d$, NCC predicts $\widehat{y}^{\mathrm{NCC}}(z) := \arg\min_{c \in \mathcal{C}} \|z - \widehat{\mu}_c\|_2^2 = \arg\max_{c \in \mathcal{C}} \{ 2 \widehat{\mu}_c^\top z - \|\widehat{\mu}_c\|_2^2 \}$, with an arbitrary but fixed tie-breaking rule.

Define the class-to-class error probability $p_{i \to j}^{\triangle} := \Pr \big( \widehat{y}^{\triangle}(z_i) = j \big)$ for $i \neq j$, where the probability is over the draw of $\widehat{S}$ and the fresh test point $x_i$. A classwise union bound yields $\mathrm{err}_{m,\mathcal{C}}^{\triangle}(f) \leq \frac{1}{|\mathcal{C}|} \sum_{i \in \mathcal{C}} \sum_{j \in \mathcal{C}, j \neq i} p_{i \to j}^{\triangle}$. Since NCC is a particular linear rule, it follows that $\mathrm{err}_{m,\mathcal{C}}^{\mathrm{LP}}(f) \leq \mathrm{err}_{m,\mathcal{C}}^{\mathrm{NCC}}(f)$. To control $p_{i \to j}^{\mathrm{NCC}}$, introduce the *pairwise margin* $\Delta_{i \to j} := \|z_i - \widehat{\mu}_j\|_2^2 - \|z_i - \widehat{\mu}_i\|_2^2$ and $p_{i \to j}^{\mathrm{NCC}} = \Pr(\Delta_{i \to j} \leq 0)$. Let $d_{ij} := \|\mu_i - \mu_j\|_2$ and $v_c := \mathbb{E} \big[ \|f(x) - \mu_c\|_2^2 \,\big|\, x \sim D_c \big]$ denote the class variances. A direct expansion with $\mathbb{E}[\delta_c] = 0$ gives $\mathbb{E}[\Delta_{i \to j}] =$

[1] Any internal randomness of the training procedure is included in the expectation below.

$d_{ij}^2 + \frac{v_j - v_i}{m}$, so larger between-class separation $d_{ij}$ and larger $m$ increase the expected margin, while variance asymmetry contributes the $\frac{v_j - v_i}{m}$ term. For simplicity, we assume throughout that $\mathbb{E}[\Delta_{i \to j}] > 0$.

## 4. Theoretical Analysis

### 4.1. Background

Above we described when a representation is favorable for few-shot adaptation; we now ask *why* a representation yields small error. Classical analyses tie few-shot performance to the clustering geometry of class embeddings.

**Supervised case.** Prior results (Galanti et al., 2022b;a; 2023b) bound these errors by the class-distance-normalized variance (CDNV). Writing $v_i := \mathbb{E}_{x \sim D_i} \|f(x) - \mu_i\|_2^2$ and $V_{ij} := V_f(D_i, D_j) := \frac{v_i + v_j}{\|\mu_i - \mu_j\|_2^2}$, one obtains (see, e.g., Prop. 7 in (Galanti et al., 2023b))

$$\mathrm{err}_{m,\mathcal{C}}^{\mathrm{LP}}(f) \leq \mathrm{err}_{m,\mathcal{C}}^{\mathrm{NCC}}(f) \lesssim \frac{1 + \frac{1}{m}}{C'} \sum_{\substack{i \in \mathcal{C} \\ j \in \mathcal{C}, \, j \neq i}} V_{ij}. \quad (1)$$

In supervised learning, representations tend to become tightly clustered, CDNV decreases, and (1) tightens (Papyan et al., 2020; Galanti et al., 2022b; 2023a; Zhou et al., 2022b). In self-supervised learning, by contrast, there is no direct pressure to reduce *total* within-class variance, so the average CDNV need not be small even when features are well organized along class-separating directions.

Under supervised training, late-layer features often exhibit *neural collapse* (NC): (i) within-class variance vanishes ($v_i \downarrow 0$); (ii) class means are centered and have equal norm; (iii) the means are approximately a simplex ETF, i.e., $\mu_i^\top \mu_j = r^2$ if $i = j$ and $\mu_i^\top \mu_j = -\frac{r^2}{C-1}$ otherwise (up to rotation/scale, with $r^2 = \|\mu_i\|_2^2$ independent of $i$); and (iv) the final linear weights align with the means. Hence inter-class gaps $d_{ij}^2 = \|\mu_i - \mu_j\|_2^2$ stay uniformly large while $v_i$ collapses, so $V_{ij} = \frac{v_i + v_j}{d_{ij}^2}$ shrinks and the CDNV average in (1) becomes small. This tightens the NCC bound (and thus LP), and explains why LP and NCC are often similar late in supervised training (Papyan et al., 2020; Galanti et al., 2022b; 2023a; Zhou et al., 2022b).

**Self-supervised case.** In self-supervised pretraining, within-class variability can remain large in nuisance directions (e.g., augmentations) while variance *along* class-separating directions is small. Classical CDNV averages over all directions and can therefore be pessimistic in such anisotropic regimes. Following Luthra et al. (2025b), we use *directional CDNV*, which retains only the within-class variance that can flip a pairwise decision.

Let $z = f(x)$ for $x \sim D_c$, with class mean $\mu_c$ and covariance $\Sigma_c$. For classes $(i, j)$, define $d_{ij} = \|\mu_j - \mu_i\|_2$ and

the decision axis $u_{ij} = (\mu_j - \mu_i)/d_{ij}$. The *directional CDNV* is $\tilde{V}_{ij} := \frac{u_{ij}^\top \Sigma_i u_{ij}}{d_{ij}^2}$, which measures class-$i$ variability *along* the one-dimensional decision axis relative to the mean gap; variance orthogonal to $u_{ij}$ does not affect the pairwise margin. We also use averages $\tilde{V}_f = \text{Avg}_{i \neq j} \tilde{V}_{ij}$, $V_f = \text{Avg}_{i \neq j} V_{ij}$, and $V_f^s = \text{Avg}_{i \neq j} \sqrt{V_{ij}}$.

Our anisotropic bounds replace the coarse dependence on $V_{ij}$ in (1) with a leading term governed by $\tilde{V}_{ij}$, plus finite-shot corrections that depend on the global dispersion and decay with the number of shots $m$. Since linear probes optimize over linear rules, $\text{err}_{m,\mathcal{C}}^{\text{LP}}(f) \leq \text{err}_{m,\mathcal{C}}^{\text{NCC}}(f)$. Concretely, Luthra et al. (2025b) showed that for any $a \geq 5$,

$$\text{err}_{m,\mathcal{C}}^{\text{NCC}}(f) \leq (C' - 1)\left[ \left(\tfrac{1}{2} - \tfrac{2}{a} - \tfrac{2^{3/2}}{am}\right)^{-2} \tilde{V}_f \right.$$
$$\left. + \tfrac{a}{4}\left(\tfrac{2}{\sqrt{m}}(V_f^s + V_f) + \tfrac{1}{m}V_f\right) \right].$$

Namely, NCC error decomposes into a decision-axis term and finite-shot leakage terms from estimating centroids (scaling with $V_f^s$ and $V_f$ at rates $m^{-1/2}$ and $m^{-1}$). Choosing $a = 16$ yields a bound of $(C' - 1)(8\,\tilde{V}_f + \frac{8}{\sqrt{m}}V_f^s + [\frac{8}{\sqrt{m}} + \frac{4}{m}]V_f)$, making explicit that in anisotropic SSL regimes the dominant contribution is governed by $\tilde{V}_f$, while the remaining terms vanish as $m$ grows.

## 4.2. Results

While the directional CDNV of (Luthra et al., 2025b) correctly targets decision-axis variability, existing guarantees can be loose or vacuous at practical shot sizes (e.g., $m \in [1, 500]$) and often entangle (i) the genuinely discriminative decision-axis term with (ii) finite-shot effects from estimating class centroids and (iii) tail behavior. Empirically, SSL embeddings are frequently *anisotropic*: much of the within-class energy can lie in subspaces nearly orthogonal to the separating directions, motivating decorrelation/whitening objectives such as Barlow Twins and VICReg and alignment–uniformity analyses in contrastive representations (Zbontar et al., 2021; Bardes et al., 2022; Wang & Isola, 2020; Wang & Liu, 2021; Chen et al., 2021).

To obtain a usable certificate in anisotropic regimes, we analyze the *pairwise margin* $\Delta_{i \to j}$ and derive bounds that (i) isolate the decision-axis second moment with an *optimal* leading constant, (ii) make finite-shot effects from empirical centroids explicit, and (iii) quantify heavy tails via a fourth-moment correction. Concretely, we introduce $\Theta_{ij} := \frac{M_{4,i} + M_{4,j}}{d_{ij}^4}$, where $M_{4,i} := \mathbb{E}\|f(x) - \mu_i\|^4$, so that tail effects appear at order $m^{-3}$, while the remaining finite-shot leakage is controlled by $V_{ij}$ and $V_{ij}^2$ at orders $m^{-1}$ and $m^{-2}$. The result below combines the pairwise and multiclass guarantees, includes a mild variance-imbalance factor

$\left(1 + \frac{v_j - v_i}{m\,d_{ij}^2}\right)^2$, and uses optimized coefficients.

**Theorem 4.1.** *Let $C' \geq 2$ and $m \geq 10$ be integers. Fix a feature map $f : \mathcal{X} \to \mathbb{R}^d$ and class-conditional distributions $D_1, \ldots, D_{C'}$ over $\mathcal{X}$. Define $E_{ij}^1 := \frac{4}{m}(V_{ij}^2 + \frac{1}{4}V_{ij})$, $E_{ij}^2 := \frac{V_{ij}}{m}$, $E_{ij}^3 := \frac{\Theta_{ij} + 2(m-1)V_{ij}^2}{m^3}$. Then the average multiclass error of the NCC classifier satisfies*

$$\text{err}_{m,\mathcal{C}}^{\text{NCC}}(f) \leq \frac{1}{C'} \sum_{i=1}^{C'} \sum_{j \neq i} \frac{4\,\tilde{V}_{ij}}{\left(1 + \frac{v_j - v_i}{m\,d_{ij}^2}\right)^2}$$
$$+ \frac{1}{C'} \sum_{i=1}^{C'} \sum_{j \neq i} \frac{\left(\sqrt{E_{ij}^1} + \sqrt{E_{ij}^2} + \sqrt{E_{ij}^3}\right)^2}{\left(1 + \frac{v_j - v_i}{m\,d_{ij}^2}\right)^2}.$$

The leading term $\frac{4\,\tilde{V}_{ij}}{\left(1 + \frac{v_j - v_i}{m\,d_{ij}^2}\right)^2}$ is the *decision-axis* contribution, governed by the within-class variance of class $i$ along the separating direction $u_{ij}$. The remaining term aggregates all *finite-shot* effects through $\frac{\left(\sqrt{E_{ij}^1} + \sqrt{E_{ij}^2} + \sqrt{E_{ij}^3}\right)^2}{\left(1 + \frac{v_j - v_i}{m\,d_{ij}^2}\right)^2}$: $E_{ij}^2$ captures linear centroid-estimation leakage ($\asymp V_{ij}/m$), $E_{ij}^1$ captures quadratic leakage ($\asymp V_{ij}^2/m$), and $E_{ij}^3$ captures tails and higher-order terms ($\asymp \Theta_{ij}/m^3 + V_{ij}^2/m^2$, since $\frac{2(m-1)}{m^3} = \mathcal{O}(m^{-2})$).

In the large-shot limit $m \to \infty$, and under mild variance balance ($v_i \approx v_j$ so $\left(1 + \frac{v_j - v_i}{m\,d_{ij}^2}\right)^2 \to 1$), the correction vanishes and the bound approaches the directional certificate $p_{i \to j}^{\text{NCC}} \lesssim 4\tilde{V}_{ij}$ (and the multiclass bound follows by union bounding and averaging over $i$). The constant 4 is optimal under second-moment information: in the known-centroid reduction the error is a one-sided tail event of the axis projection with variance $d_{ij}^2 \tilde{V}_{ij}$, and Cantelli's inequality yields the minimax-tight factor 4 (App. D). Finally, $\left(1 + \frac{v_j - v_i}{m\,d_{ij}^2}\right)^2 = \left(1 + \frac{v_j - v_i}{m\,d_{ij}^2}\right)^2$ is a mild multiplicative perturbation whenever $\frac{|v_j - v_i|}{m\,d_{ij}^2}$ is not too large, and it disappears as $m$ grows.

**Optimality of the leading constant.** The coefficient 4 in the leading decision-axis term is not an artifact of our proof technique. In the idealized known-centroid limit ($m \to \infty$), the pairwise NCC error $p_{i \to j}^{\text{NCC}}$ reduces to a one-dimensional tail event along the separating axis: if $u_{ij} := (\mu_j - \mu_i)/\|\mu_j - \mu_i\|_2$ and $X := (z_i - \mu_i)^\top u_{ij}$ for $z_i \sim D_i$, then $p_{i \to j}^{\text{NCC}} = \Pr\left(X \geq \|\mu_j - \mu_i\|_2/2\right)$ and $\text{Var}(X) = \|\mu_j - \mu_i\|_2^2\,\tilde{V}_{ij}$. Among all mean-zero random variables with this variance, Cantelli's inequality yields the sharp distribution-free bound $p_{i \to j}^{\text{NCC}} \leq \frac{4\tilde{V}_{ij}}{1 + 4\tilde{V}_{ij}} \leq 4\tilde{V}_{ij}$, and the constant is minimax-tight via a two-point construction. Consequently, any universally valid second-moment bound of the form $p_{i \to j}^{\text{NCC}} \leq c\,\tilde{V}_{ij} + (\text{lower-order terms})$ must have $c \geq 4$ unless additional tail assumptions are imposed (e.g.,

subgaussianity). See App. D for details and the connection to the finite-shot factor $1/\left(1 + \frac{v_j - v_i}{m\, d_{ij}^2}\right)^2$.

### 4.3. Near-orthogonality from small directional CDNV

A single SSL representation can support many downstream tasks even when the *total* within-class spread is large. The reason is that few-shot error (and our bounds) are controlled by variability *along the decision axis*, not by variability in directions that are irrelevant to the margin. Directional CDNV measures within-class variance projected onto the class-separating direction, whereas classical CDNV sums variance over all directions and can therefore be much larger.

We make two complementary points. (i) For two *independent* balanced binary labelings, small directional CDNV for both tasks implies their decision axes are nearly orthogonal. (ii) There exist natural representation families where directional CDNV is simultaneously small for many tasks while CDNV is large, because most within-class variability lies in directions orthogonal to every task's decision axis.

**Setup: two independent balanced multiclass labelings.**
Let $z = f(x) \in \mathbb{R}^d$ with $\mathbb{E}\|z\|_2^2 < \infty$, and let $y^{(1)} \in [K_1]$, $y^{(2)} \in [K_2]$ be two labelings of the same examples (where $[K] := \{1, \dots, K\}$). Assume both tasks are balanced and independent, i.e., for all $a \in [K_1]$ and $b \in [K_2]$, we have

$$\Pr(y^{(1)} = a) = \frac{1}{K_1}, \quad \Pr(y^{(2)} = b) = \frac{1}{K_2}, \quad y^{(1)} \perp y^{(2)}.$$

For each task $\ell \in \{1, 2\}$ and class index $c \in [K_\ell]$, define $\mu_c^{(\ell)} := \mathbb{E}[z \mid y^{(\ell)} = c]$, and $\Sigma_c^{(\ell)} := \mathrm{Cov}(z \mid y^{(\ell)} = c)$. For any pair of distinct classes $a \neq a'$ in task 1, define the pairwise mean gap and (unit) decision axis $d_{aa'}^{(1)} := \|\mu_a^{(1)} - \mu_{a'}^{(1)}\|_2$ and $u_{aa'}^{(1)} := \frac{\mu_a^{(1)} - \mu_{a'}^{(1)}}{d_{aa'}^{(1)}}$, and similarly, for any $b \neq b'$ in task 2, $d_{bb'}^{(2)} := \|\mu_b^{(2)} - \mu_{b'}^{(2)}\|_2$ and $u_{bb'}^{(2)} := \frac{\mu_b^{(2)} - \mu_{b'}^{(2)}}{d_{bb'}^{(2)}}$ (under the assume that $d_{aa'}^{(1)} > 0$ and $d_{bb'}^{(2)} > 0$).

For a fixed pair $(a, a')$ in task 1, define the directional CDNV (maximized over all classes of task 1) by

$$\tilde{V}_{aa',c}^{(1)} := \frac{\left(u_{aa'}^{(1)}\right)^\top \Sigma_c^{(1)} u_{aa'}^{(1)}}{\left(d_{aa'}^{(1)}\right)^2}, \quad \tilde{V}_{aa'}^{(1)} := \max_{c \in [K_1]} \tilde{V}_{aa',c}^{(1)},$$

and analogously, for a fixed pair $(b, b')$ in task 2, we define $\tilde{V}_{bb',c}^{(2)}$ and $\tilde{V}_{bb'}^{(2)}$

(For reference, the classical pairwise CDNV for task 1 and pair $(a, a')$ is

$$V_{aa'}^{(1)} := \frac{\mathrm{tr}(\Sigma_a^{(1)}) + \mathrm{tr}(\Sigma_{a'}^{(1)})}{\left(d_{aa'}^{(1)}\right)^2},$$

and similarly for task 2.)

**Small directional CDNV forces near-orthogonality (multiclass version).** The next proposition shows that small directional CDNV for two *independent* multiclass tasks yields small overlap between any pairwise decision axes across the two tasks.

**Proposition 4.2** (Near-orthogonality from small directional CDNV). *Assume $y^{(1)} \in [K_1]$ and $y^{(2)} \in [K_2]$ are balanced and independent. Fix any $a \neq a'$ in $[K_1]$ and any $b \neq b'$ in $[K_2]$, and assume $d_{aa'}^{(1)}, d_{bb'}^{(2)} > 0$. Then*

$$\left| \left(u_{aa'}^{(1)}\right)^\top u_{bb'}^{(2)} \right| \leq \min \left\{ \frac{d_{aa'}^{(1)}}{d_{bb'}^{(2)}} \sqrt{2 K_2 \tilde{V}_{aa'}^{(1)}}, \ \frac{d_{bb'}^{(2)}}{d_{aa'}^{(1)}} \sqrt{2 K_1 \tilde{V}_{bb'}^{(2)}} \right\}.$$

Prop. 4.2 shows that for two *independent* balanced tasks, small directional CDNV implies that *any* pairwise class-separating direction from task 1 has small overlap with *any* pairwise class-separating direction from task 2. In other words, when within-class variability is small along the relevant margin direction for each task, the representation cannot reuse the same direction to separate classes for another independent task, except for a small residual overlap controlled by directional CDNV.

A useful consequence is that a single representation can support many independent tasks by allocating nearly orthogonal discriminative directions to different tasks, while still allowing substantial total within-class variance in directions that are orthogonal to these decision axes. This is precisely the regime where directional CDNV can be small for many tasks simultaneously, even if the classical CDNV (which sums variance across all directions) is large.

**Why CDNV can still be large.** Fix a task $\ell \in \{1, 2\}$ and a pair of classes $c \neq c' \in [K_\ell]$, and write $u := u_{cc'}^{(\ell)}, d := d_{cc'}^{(\ell)}$, and $\Pi := I - uu^\top$. For any class $r \in [K_\ell]$, the covariance decomposes as $\Sigma_r^{(\ell)} = (uu^\top)\Sigma_r^{(\ell)}(uu^\top) + (uu^\top)\Sigma_r^{(\ell)}\Pi + \Pi\Sigma_r^{(\ell)}(uu^\top) + \Pi\Sigma_r^{(\ell)}\Pi$. The directional CDNV for the pair $(c, c')$ only depends on the on-axis variance, namely $\tilde{V}_{cc',r}^{(\ell)} = \frac{u^\top \Sigma_r^{(\ell)} u}{d^2}$ and $\tilde{V}_{cc'}^{(\ell)} = \max_{r \in [K_\ell]} \tilde{V}_{cc',r}^{(\ell)}$, whereas the classical pairwise CDNV $V_{cc'}^{(\ell)} = \frac{\mathrm{tr}(\Sigma_c^{(\ell)}) + \mathrm{tr}(\Sigma_{c'}^{(\ell)})}{d^2}$ counts variance in all directions. In particular, since $\mathrm{tr}(\Sigma_r^{(\ell)}) = u^\top \Sigma_r^{(\ell)} u + \mathrm{tr}(\Pi\Sigma_r^{(\ell)}\Pi)$, the off-axis contribution $\mathrm{tr}(\Pi\Sigma_r^{(\ell)}\Pi)$ can be made arbitrarily large without changing $\tilde{V}_{cc',r}^{(\ell)}$ (and hence without changing $\tilde{V}_{cc'}^{(\ell)}$).

**Many tasks with small dir-CDNV but large CDNV.** To illustrate the phenomenon transparently, consider $M \leq d$ independent balanced binary tasks (a special case of the multiclass setup). Let $v_1, \dots, v_M \in \mathbb{R}^d$ be orthonormal, let $t^{(1)}, \dots, t^{(M)} \in \{\pm 1\}$ be i.i.d. balanced, and define $z = \sum_{\ell=1}^M \frac{\Delta_\ell}{2} t^{(\ell)} v_\ell + \eta + \xi$, where $\eta \in \mathrm{span}\{v_1, \dots, v_M\}^\perp$ and $(\eta, \xi)$ are independent of $(t^{(1)}, \dots, t^{(M)})$.

For task $\ell$, taking conditional expectations gives $\mu_+^{(\ell)} -$

$\mu_-^{(\ell)} = \Delta_\ell v_\ell$, so $v_\ell$ is exactly the decision axis and the pairwise mean gap is $\Delta_\ell$. Conditioning on $t^{(\ell)}$ does not affect the distribution of $\eta$, $\xi$, or $\{t^{(j)} : j \neq \ell\}$. Using $v_\ell^\top \eta = 0$ and $v_\ell^\top v_j = 0$ for $j \neq \ell$, we get $v_\ell^\top \Sigma_s^{(\ell)} v_\ell = v_\ell^\top \text{Cov}(\xi) v_\ell$ for every $s \in \{\pm 1\}$, and therefore $\tilde{V}^{(\ell)} = \frac{v_\ell^\top \text{Cov}(\xi) v_\ell}{\Delta_\ell^2}$. Thus, if $\text{Cov}(\xi)$ is small on $\mathcal{U} := \text{span}\{v_1, \dots, v_M\}$, then $\tilde{V}^{(\ell)}$ is small for every $\ell$.

By contrast, classical CDNV aggregates variance in all directions and therefore also counts the off-axis nuisance energy: $V^{(\ell)} = \frac{\text{tr}(\Sigma_+^{(\ell)}) + \text{tr}(\Sigma_-^{(\ell)})}{\Delta_\ell^2} \geq \frac{2 \, \text{tr}(\text{Cov}(\eta))}{\Delta_\ell^2}$. This can be made arbitrarily large by increasing $\text{tr}(\text{Cov}(\eta))$ while keeping all directional quantities fixed.

This example shows that an orthogonal-factor structure is a convenient *sufficient* mechanism for obtaining small directional CDNV across many labelings, but it is not required for Proposition 4.2. The near-orthogonality conclusion there follows directly from task independence and small variance along the relevant pairwise decision axes.

# 5. Experiments

## 5.1. Settings

**Datasets and augmentations.** We conduct our experiments on the standard image classification dataset – mini-ImageNet (Vinyals et al., 2016) which is a subset of ImageNet-1K (Deng et al., 2009), CelebA (Liu et al., 2015) and CUB-200 (Wah et al., 2011) in the main text. Please refer to App. B for additional experiments on ImageNet-1K (Deng et al., 2009) and SVHN (Netzer et al., 2011).

**Methods, optimizers and augmentations.** We study a range of self-supervised paradigms, including contrastive learning (SimCLR (Chen et al., 2020), VICReg (Bardes et al., 2022)), masked modeling (I-JEPA (Assran et al., 2023), MAE (He et al., 2022)), distillation-based methods (DINO-v2 (Oquab et al., 2024)), and multimodal pretraining (CLIP (Radford et al., 2021), and SigLIP (Zhai et al., 2023). For learning-dynamics experiments, we train SimCLR, VICReg, MAE, and DINO-v2 from scratch on mini-ImageNet. Across experiments, we use ResNet-18/ResNet-50 backbones (He et al., 2016) (with width multiplier 2) and a ViT-Base (ViT-B/16) backbone (Dosovitskiy et al., 2021), and follow the standard augmentation pipelines associated with each method. Please refer to App. A for the full training, optimization, and augmentation details.

## 5.2. Results

**Directional-CDNV vs. CDNV.** In Fig. 2 we test whether SSL training preferentially minimizes directional CDNV. To do so, we pretrain MAE, SimCLR, DINO-v2, and VICReg for 1000 epochs and track both standard CDNV and

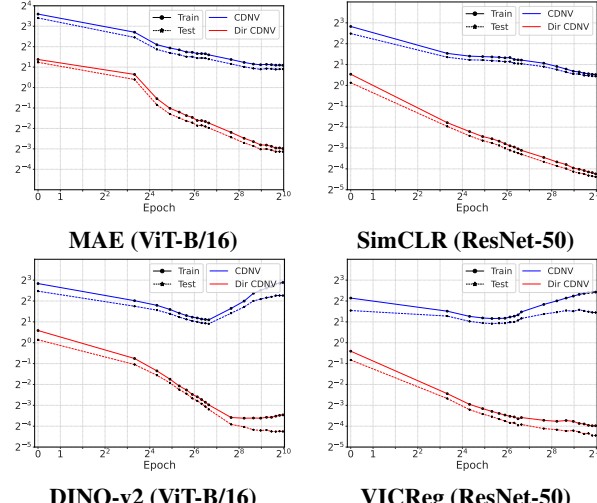

**MAE (ViT-B/16)**     **SimCLR (ResNet-50)**

**DINO-v2 (ViT-B/16)**     **VICReg (ResNet-50)**

*Figure 2.* **Decision-axis collapse emerges during SSL training.** We track both CDNV and directional CDNV on the training and test sets. Directional CDNV decreases much more than CDNV, indicating that SSL primarily tightens class geometry along separating directions even when overall within-class variability is large.

directional CDNV on the training and test sets. As shown in Fig. 2, directional CDNV drops dramatically over training, from roughly $2^{-1}-2^1$ down to about $2^{-3}-2^{-5}$. In contrast, standard CDNV decreases only modestly and in some cases even increases transiently. These trends suggest that full within-class variance collapse is not the dominant effect in SSL. Instead, SSL mainly suppresses variability along class-separating directions, and this directional form of collapse appears consistently across a broad range of methods. Refer to Fig. 7 (top) for additional results with large models and bigger datasets.

**Validating Thm. 4.1 with varying $m$.** We use frozen, off-the-shelf vision encoders, pretrained on IM-1K for vision-only SSL methods, on 400M image-text pairs (Radford et al., 2021) for CLIP, and on WebLI (Chen et al., 2023) for SigLIP. We then evaluate nearest-class-centroid (NCC) classification on mini-ImageNet without finetuning, using binary tasks ($C = 2$) formed by randomly selecting two classes. For each shot count $m$, we estimate class centroids from $m$ randomly sampled training examples per class and report test accuracy; we average over both the random choice of the two classes and the random draw of the $m$ shots, repeating this procedure five times.

As shown in Fig. 3, our finite-$m$ certificates are informative at practical shot counts and closely track the observed few-shot error. The finite-$m$ curve converges (by construction) to the directional-only $m \to \infty$ limit shown in the figure. Importantly, the bound is non-vacuous in this regime: for moderate $m$, our certified error drops below the $0.5$ chance threshold, whereas the bound of (Luthra et al., 2025b) re-

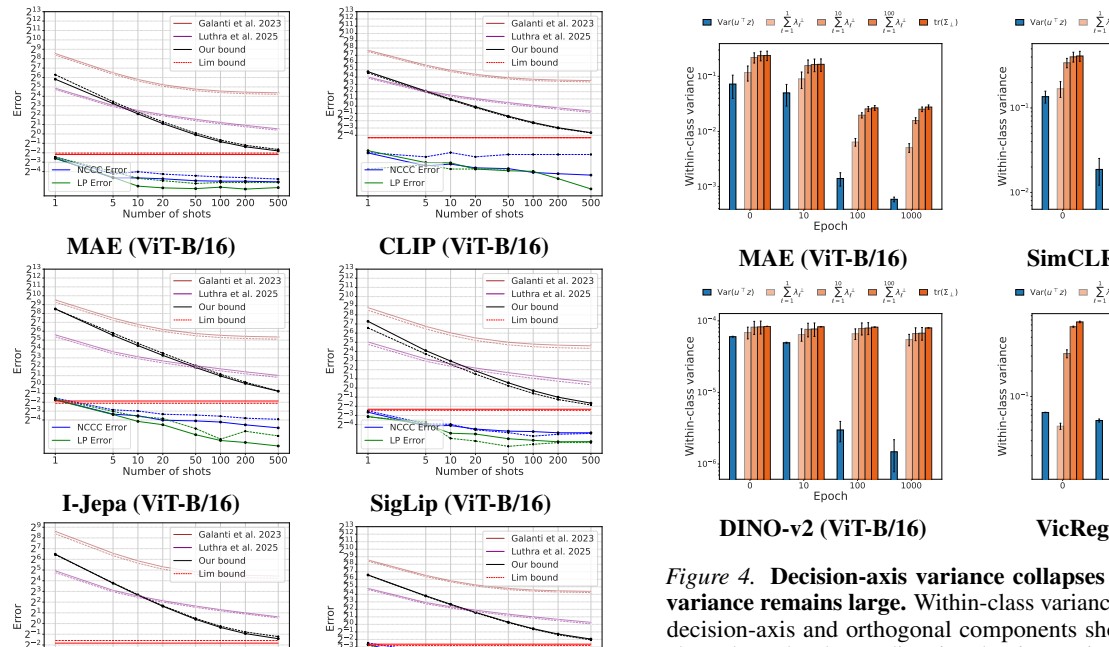

**MAE (ViT-B/16)**     **CLIP (ViT-B/16)**

**I-Jepa (ViT-B/16)**     **SigLip (ViT-B/16)**

**SimCLR (ResNet-50)**     **VICReg (ResNet-50)**

*Figure 3.* **Decision-axis variance yields informative few-shot certificates in SSL.** We plot few-shot NCC and few-shot LP test error versus the number of shots per class, $m$, for several pretrained SSL encoders, together with certified upper bounds from our analysis. We compare our finite-$m$ bound to the directional-only $m \to \infty$ limit and to the bounds of (Luthra et al., 2025b; Galanti et al., 2023b).

**MAE (ViT-B/16)**     **SimCLR (ResNet-50)**

**DINO-v2 (ViT-B/16)**     **VicReg (ResNet-50)**

*Figure 4.* **Decision-axis variance collapses while orthogonal variance remains large.** Within-class variance decomposed into decision-axis and orthogonal components shows rapid collapse along the task-relevant direction despite persistently large orthogonal variance.

mains above $0.5$ here and is therefore vacuous.

To further validate our proposed bound, we evaluate Linear Probing (LP) accuracy which is a more commonly used metric for measuring downstream performance in SSL. We also include comparison against prior generalization bounds (Galanti et al., 2023b) and show that our bound remains tighter in few-shot setting.

**Decision-axis vs orthogonal variance.** To examine how within-class variance evolves during self-supervised training, we analyze the geometry of learned representations at the level of class pairs. For each model and training epoch, we randomly select 20 class pairs $(i, j)$ and define the corresponding decision axis $u_{ij}$ as the normalized difference between class means. Using this axis and the within-class covariance matrix of representations $\Sigma_{ij} = \frac{1}{N_{ij}} \sum_{n:\, y_n \in \{i,j\}} (z_n - \mu_{y_n})(z_n - \mu_{y_n})^\top$, we decompose it into a component along the decision axis as $u_{ij}^\top \Sigma_{ij} u_{ij}$. We compute the corresponding orthogonal covariance matrix as $P_{ij}^\perp \Sigma_{ij} P_{ij}^\perp$ where $P_{ij}^\perp = I - u_{ij} u_{ij}^\top$. We then perform an eigen-decomposition of the orthogo-

nal covariance matrix and compute the cumulative variance captured by its top $k$ principle directions.

In Fig. 4, we report variance along the decision axis, cumulative orthogonal variance for $k \in \{1, 10, 100\}$, and the total orthogonal variance, averaged across the selected pairs. Across all methods, variance along the decision axis collapses rapidly with training, while substantial variance persists in the orthogonal directions which are irrelevant for task-specific downstream classification. Please refer to Fig. 7 (bottom) for additional results with large models and bigger datasets.

**Multitask orthogonal representation decomposition.** To study how representations learned via self-supervised learning organize multiple semantic labelings in high-dimensional feature space, we design a controlled synthetic experiment (Fig. 5). Each sample is a $64 \times 64$ image generated from a combination of independent factors of variation, including *color* ($L_1$), *background style* ($L_2$), *shape* ($L_3$), and *shape size* ($L_4$), yielding multiple valid labelings over the same set of inputs. We pre-train a ResNet-18 encoder from scratch on this dataset using self-supervised objectives. At different stages of training, we extract the learned feature representations and compute class means for each labeling $L_n$, denoted by $\mu_y^{(L_n)}$. For each labeling, we define *decision axes* as normalized differences between class means: $u_{y_i y_j}^{(L_n)} = (\mu_{y_i}^{(L_n)} - \mu_{y_j}^{(L_n)}) / \|\mu_{y_i}^{(L_n)} - \mu_{y_j}^{(L_n)}\|$, where $y_i \neq y_j$. These directions characterize the linear decision geometry associated with each labeling. To quantify the interaction between different labelings, we compute the cosine simi-

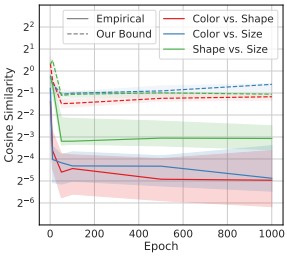 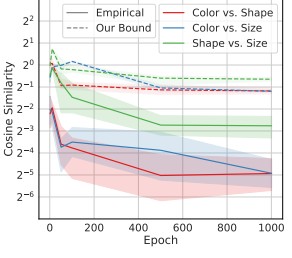

**(a) SimCLR (ResNet-18)**    **(b) VicReg (ResNet-50)**

*Figure 5.* **Multitask decision-axis orthogonalization in SSL.** Median absolute cosine similarity (25–75% bands) between decision axes from different semantic labelings during training; alignments decay toward zero and are upper-bounded by our theory (dashed).

larity between decision axes drawn from *distinct* labelings, i.e., $|\langle u_{ij}^{(L_a)}, u_{kl}^{(L_b)} \rangle|$ for $L_a \neq L_b$, and aggregate statistics across all class pairs.

In Fig. 5, we report the median absolute cosine similarity together with interquartile (25–75%) bands over training. At random initialization, decision axes corresponding to different labelings exhibit substantial alignment, indicating entangled representations. As training progresses, these cosine similarities rapidly decrease and stabilize near zero, demonstrating that the learned representation decomposes into *approximately orthogonal semantic subspaces* corresponding to different labelings. Importantly, this orthogonalization occurs simultaneously for multiple independent labelings, indicating that SSL representations do not collapse to a single discriminative direction but instead support multiple, low-interference decision geometries.

**Multitask Geometry on Real Data.** We extend our analysis to real datasets in Fig. 6. We train a VICReg with ResNet-50 backbone on CelebA and CUB-200, and validate our claims of multi-task orthogonality on selected attributes. Similar to our synthetic experiments, we tracked the pairwise cosine similarity between selected attributes and evaluated our theoretical bound from Prop. 4.2 across training epochs. Tracking this progression illustrates how the representations naturally restructure from a random initialization into task-specific orthogonal subspaces, strongly reinforcing our theoretical arguments. Additionally, we further investigate geometric structure of representations learned by large foundation models in App. B.3.

## 6. Conclusion

Our work builds on recent progress in understanding why self-supervised representations transfer well, and strengthens the case that directional collapse is central to this phenomenon. In particular, we solidify that few-shot transfer is governed not by global within-class variance, but by vari-

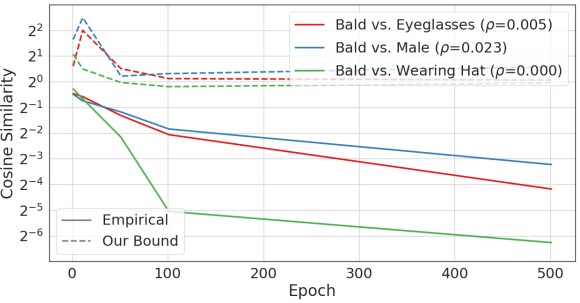

**(a) CelebA**

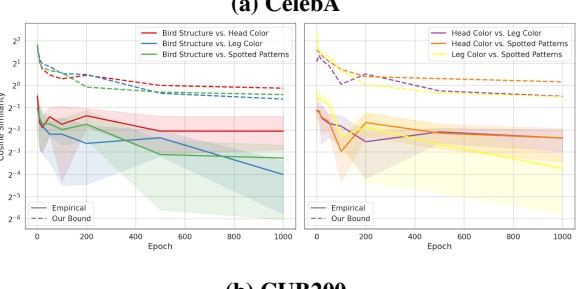

**(b) CUB200**

*Figure 6.* **Multitask decision-axis orthogonalization in SSL representations on real datasets.** Our bound in Prop. 4.2 vs. cosine similarity between decision axes induced by different semantic tasks decreases throughout training, indicating that self-supervised learning organizes distinct labelings into approximately orthogonal subspaces.

ance along class-separating directions.

Compared to recent work, we derive sharper non-asymptotic bounds for nearest-class-centroid classification and linear probing, in which directional CDNV appears as the leading term, together with explicit finite-shot correction terms, a separate fourth-moment contribution, and an optimal leading constant under second-moment assumptions. We also show empirically, across diverse SSL methods and architectures, that directional CDNV consistently collapses during pretraining even when classical CDNV remains large. Finally, we connect directional collapse to multitask structure by showing that small directional CDNV across independent tasks implies near-orthogonality of their decision axes, linking strong few-shot transfer to efficient packing of multiple tasks in a shared representation.

Taken together, these results provide a sharper theoretical account, broader empirical support, and a new multitask perspective on why modern SSL representations support strong few-shot transfer.

## Impact Statement

This work is theoretical and raises no direct ethical, safety, or environmental concerns. We study conditions under which self-supervised learning methods adapt to downstream tasks, introduce representation properties that help explain when this is possible, and provide empirical evidence supporting these claims. These results inform the high-level design and evaluation of future self-supervised learning algorithms, with very limited direct societal impact.

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

# A. Additional Experimental Details

In this section we describe the experimental settings. Please refer to App. A for the complete details.

**Datasets and augmentations.**     We conduct our experiments primarily on the standard image classification dataset mini-ImageNet (Vinyals et al., 2016) which is a subset of ImageNet-1K (Deng et al., 2009). It contains images at their original resolution (224×224) and has 50000 train, 10000 validation, and 5000 test images, with a total 100 of 1000 ImageNet classes. We additionally evaluate on the CelebA dataset (Liu et al., 2015), CUB-200-2011 dataset (Wah et al., 2011) and SVHN (Netzer et al., 2011).

**Methods and optimizers.**     We consider several families of self-supervised learning methods, including contrastive learning (SimCLR, VICReg), masked modeling (I-JEPA, MAE), distillation-based approaches (DINO-v2), and multimodal learning (CLIP, SigLIP). For experiments focused on analyzing learning dynamics, we train SimCLR, VICReg, MAE, and DINO-v2 models from scratch on mini-ImageNet.

For SimCLR, we use DCL loss (Yeh et al., 2022) in place of the standard InfoNCE loss and optimize the model using LARS (You et al., 2017). The momentum is set to $0.9$ and the weight decay to $1e^{-6}$. The learning rate is scaled linearly with the batch size as $0.3 \cdot \lfloor B/256 \rfloor$ (Chen et al., 2020)., and we use a batch size of $1024$ throughout training. We apply a linear warm-up (Goyal et al., 2017) for the first 10 epochs, followed by a cosine learning rate schedule without restarts (Loshchilov & Hutter, 2016) for the remainder of training.

For VicReg, we use a ResNet-50 backbone (He et al., 2016) with a 2-layer projection head (hidden and output dimensions 2048). The VICReg loss (Bardes et al., 2022) combines three objectives: invariance loss for view similarity, variance loss to prevent collapse, and covariance loss for dimension decorrelation, with coefficients $\lambda = 25.0$, $\mu = 25.0$, and $\nu = 1.0$. The model is optimized using AdamW (Loshchilov & Hutter, 2019) with weight decay 0.05. The learning rate is scaled as $0.0005 \cdot (B \times W)/256$ where $B$ is the batch size per GPU, with 10-epoch linear warm-up followed by cosine decay (Loshchilov & Hutter, 2016) to 0.0.

For MAE (He et al., 2022), we use mean-squared error between reconstructed and input image patches as our objective function. The target pixel values are normalized per-patch as it encourages the model to learn richer, contrast-invariant semantic features. We use a masking ratio of $0.75$, meaning that $75\%$ of image patches are masked and only the remaining patches are passed to the encoder. The decoder is a lightweight Vision Transformer with 8 transformer blocks, 16 attention heads, and an embedding dimension of $512$. We follow the same learning rate scheduler and scaler as SimCLR, except the base learning rate is set to $1.5e^{-4}$ (instead of $0.3$). Additionally, weight-decay for all layers is set to $0.05$ except LayerNorm and bias terms. We use AdamW (Loshchilov & Hutter, 2019) optimizer following standard MAE setup.

For DINO, we use the DINOv2 framework with a Vision Transformer Base (ViT-B/16) architecture combining three loss objectives: DINO loss (Caron et al., 2021) for global consistency, iBOT patch loss (Zhou et al., 2022a) for local feature learning with masked patches, and KoLeo loss for feature uniformity. The model is optimized using AdamW (Loshchilov & Hutter, 2019) with weight decay 0.04. The learning rate is scaled as $0.001 \cdot \lfloor (B \times W)/256 \rfloor$ (batch size 256), with 10-epoch linear warm-up followed by cosine decay (Loshchilov & Hutter, 2016) to $1 \times 10^{-5}$. The teacher network uses EMA with cosine momentum schedule (0.992 to 1.0) and temperature warm-up (0.04 to 0.07 over 30% of training). Data augmentation employs 2 global and 8 local crops with block masking (ratio 0.6) on global crops for iBOT.

**SimCLR Augmentations.**   As described in SimCLR (Chen et al., 2020), we use the following pipeline: random resized cropping to $224 \times 224$, random horizontal flipping, color jittering (brightness, contrast, saturation: 0.8; hue: 0.2), random grayscale conversion ($p = 0.2$), and Gaussian blur (applied with probability 0.1 using a $3 \times 3$ kernel and $\sigma = 1.5$).

**VicReg Augmentations.**   Following VICReg (Bardes et al., 2022), we adopt the SimCLR-style image augmentation protocol and symmetrized across views. Two random crops are sampled from each image and resized to $224 \times 224$. Each view undergoes random horizontal flipping, color jittering (brightness: 0.4, contrast: 0.4, saturation: 0.2, hue: 0.1; applied with probability 0.8), random grayscale conversion ($p = 0.2$), Gaussian blur ($p = 0.5$, kernel size 23), and solarization ($p = 0.1$). Finally, images are normalized per channel using the ImageNet statistics.

**MAE Augmentations.**     For MAE, we do not apply strong appearance-based augmentations such as color jittering, grayscale conversion, or Gaussian blurring, as these can adversely affect the pixel reconstruction objective. Instead, we use a RandomResizedCrop with *bicubic* interpolation and a scale range of $(0.2, 1.0)$, followed by normalization using ImageNet-1K statistics.

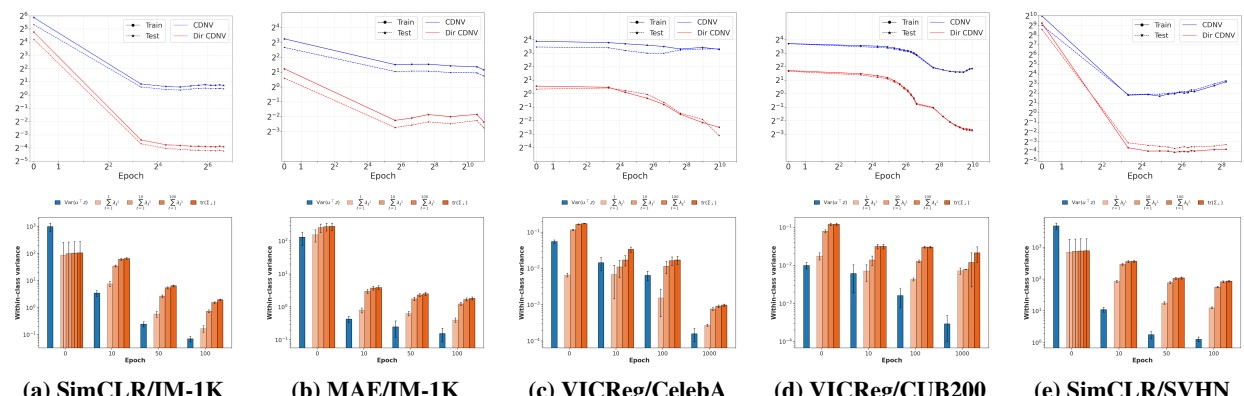

| (a) SimCLR/IM-1K | (b) MAE/IM-1K | (c) VICReg/CelebA | (d) VICReg/CUB200 | (e) SimCLR/SVHN |

*Figure 7.* **Large-scale evaluations.** **Top row:** standard CDNV and directional CDNV during SSL training. **Bottom row:** decomposition of within-class variance into decision-axis and orthogonal components. We evaluate MAE with a ViT-L/16 backbone on ImageNet-1K and SimCLR/VICReg with ResNet-50 backbones across ImageNet-1K, CelebA, CUB200, and SVHN.

**DINO-v2 Augmentations.** Following DINOv2 (Oquab et al., 2024), we use a multi-crop image augmentation strategy inherited from DINO. For each image, two global views are sampled using random resized crops and resized to $224 \times 224$, along with several local views resized to $96 \times 96$. Each view is independently augmented using random horizontal flipping, color jittering of brightness, contrast, saturation, and hue (applied with probability 0.8), and random grayscale conversion ($p = 0.2$). Gaussian blur is applied to the global views with different probabilities (strong blur on one global crop and weaker blur on the other), and solarization is applied to one of the global crops with probability 0.2.

**Backbone architectures.** We use multiple backbone architectures to demonstrate that our observations are not specific to a particular model family. Specifically, we consider ResNet-18 and ResNet-50 (He et al., 2016) encoders with a width multiplier of 2, as well as a Vision Transformer (ViT-Base) (Dosovitskiy et al., 2021).

The ViT-Base (ViT-B/16) architecture consists of 12 transformer layers, each with 12 attention heads and a hidden dimension of 768. For input images of size $224 \times 224$, we use a patch size of $16 \times 16$, yielding 196 patch tokens along with a single [CLS] token. The MLP blocks have a hidden dimension of 3072, and layer normalization is applied prior to each attention and MLP block.

For SimCLR and VicReg, the backbone encoders are followed by a projection head with a standard two-layer MLP architecture composed of: Linear(2048 → 2048) → ReLU → Linear(2048 → 128).

# B. Additional Experiments

### B.1. Scaling Directional Collapse Across Bigger Models and Datasets

In the main text, we demonstrated that directional CDNV collapses substantially during SSL training on mini-ImageNet across multiple SSL paradigms. We now extend these observations to larger architectures, larger-scale datasets, and settings with class imbalance. Across all settings, we consistently observe that SSL preferentially suppresses variance along class-separating directions while substantial orthogonal variance persists.

**Large scale models and datasets.** In Fig. 7 (a-d), we extend our observations to larger-scale models and datasets, including MAE (ViT-L/16) and SimCLR (ResNet-50) pretrained on ImageNet-1K, as well as VICReg (ResNet-50) trained on CelebA and CUB200. Across all settings, directional CDNV decreases substantially more than standard CDNV throughout training.

**Robustness under class imbalance.** We additionally evaluate directional collapse under class imbalance by training SimCLR (ResNet-50) on SVHN with imbalanced class frequencies. As shown in Fig. 7 (d), SSL continues to preferentially reduce variance along class-separating directions despite the imbalance, suggesting that directional collapse is not merely restricted to class-balanced datasets.

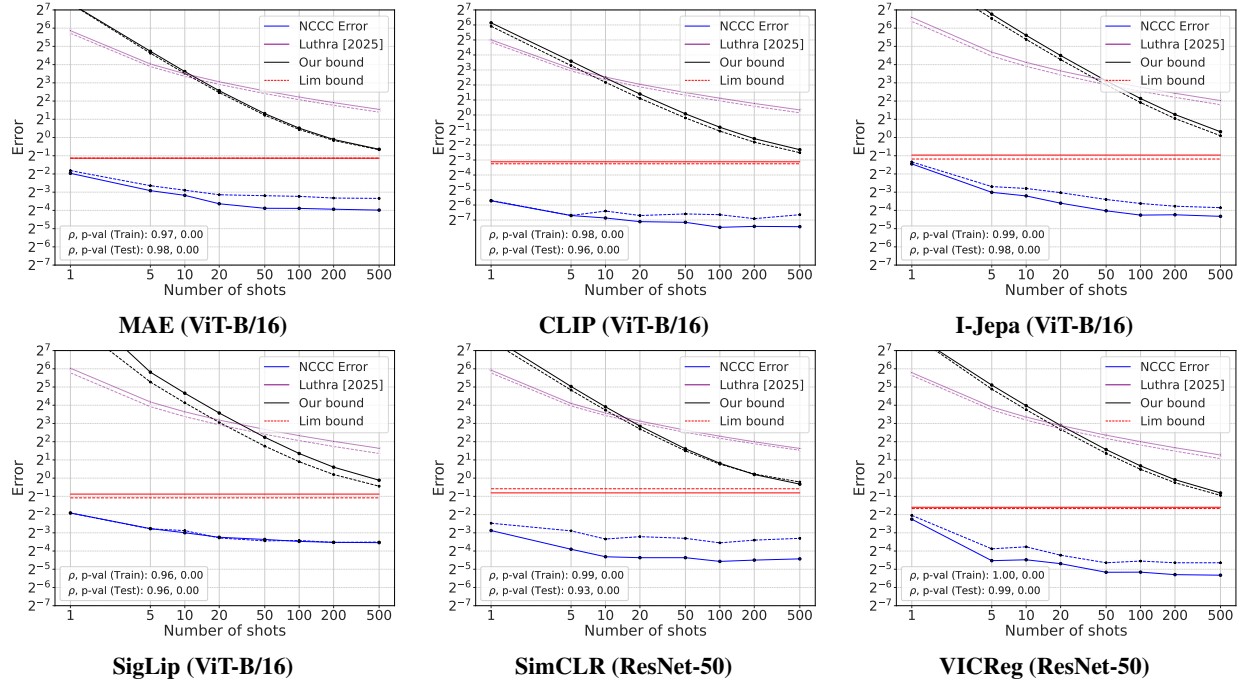

*Figure 8.* **Decision-axis variance yields reliable certificates for non-binary tasks.** Few-shot NCC error versus the number of shots per class for $C = 3$. The experimental setup is same as described for Fig. 3.

### B.2. Extended Validation of Few-Shot Bounds

**Validating Thm. 4.1 for higher number of classes.** We repeat the experiment shown in Fig. 3 with same vision encoders for $C = 3$, and show that our bound provides reliable certificates for varying $m$ in non-binary tasks. As shown in Fig. 8, our bound predicts error close to 0.5 for practical values of $m$.

### B.3. Multitask Geometry on Real Data

In the main text, we demonstrated that SSL representations progressively orthogonalize decision axes corresponding to different semantic labelings in a controlled synthetic setting and in-distribution real datasets. We now extend this analysis to large pretrained foundation models.

**Foundation model representations.** In Fig. 9, we additionally analyze pretrained CLIP and DINOv2 representations on CUB200. We again observe that pairs of tasks with low label covariance exhibit substantially smaller decision-axis alignment, indicating that multitask orthogonalization persists even in large-scale pretrained foundation models.

### B.4. Hierarchical clustering

We analyze the CDNV for self-supervised learning models using a hierarchical class structure. We generated 10 semantic superclasses for Mini-ImageNet's 100 classes using GPT 4o to group related categories, such as *birds* (house finch, robin, toucan), *dogs* (golden retriever, boxer, dalmatian), and *marine life* (jellyfish, king crab, coral reef). This allows us to analyze whether SSL models learn representations that respect semantic groupings and how CDNV evolves during training both within and across superclasses.

## C. Proofs

Assume all samples are i.i.d. within each class and independent across classes. Let $z_c = f(x_c)$ for $x_c \sim D_c$ have mean $\mu_c := \mathbb{E}[z_c]$ and covariance $\Sigma_c := \mathrm{Cov}(z_c)$ for $c \in \{i, j\}$. For $m \geq 1$ samples per class, draw support inputs $\{x_{c,s}\}_{s=1}^m$ i.i.d. from $D_c$ for $c \in \{i, j\}$, set $z_{c,s} := f(x_{c,s})$, and define $\widehat{\mu}_c := \frac{1}{m} \sum_{s=1}^m z_{c,s}$. Define the sample-mean errors $\delta_c := \widehat{\mu}_c - \mu_c$ for $c \in \{i, j\}$. Draw a test input $x_i \sim D_i$ independently of the support inputs and set $z_i := f(x_i)$. Fix $i \neq j$ with

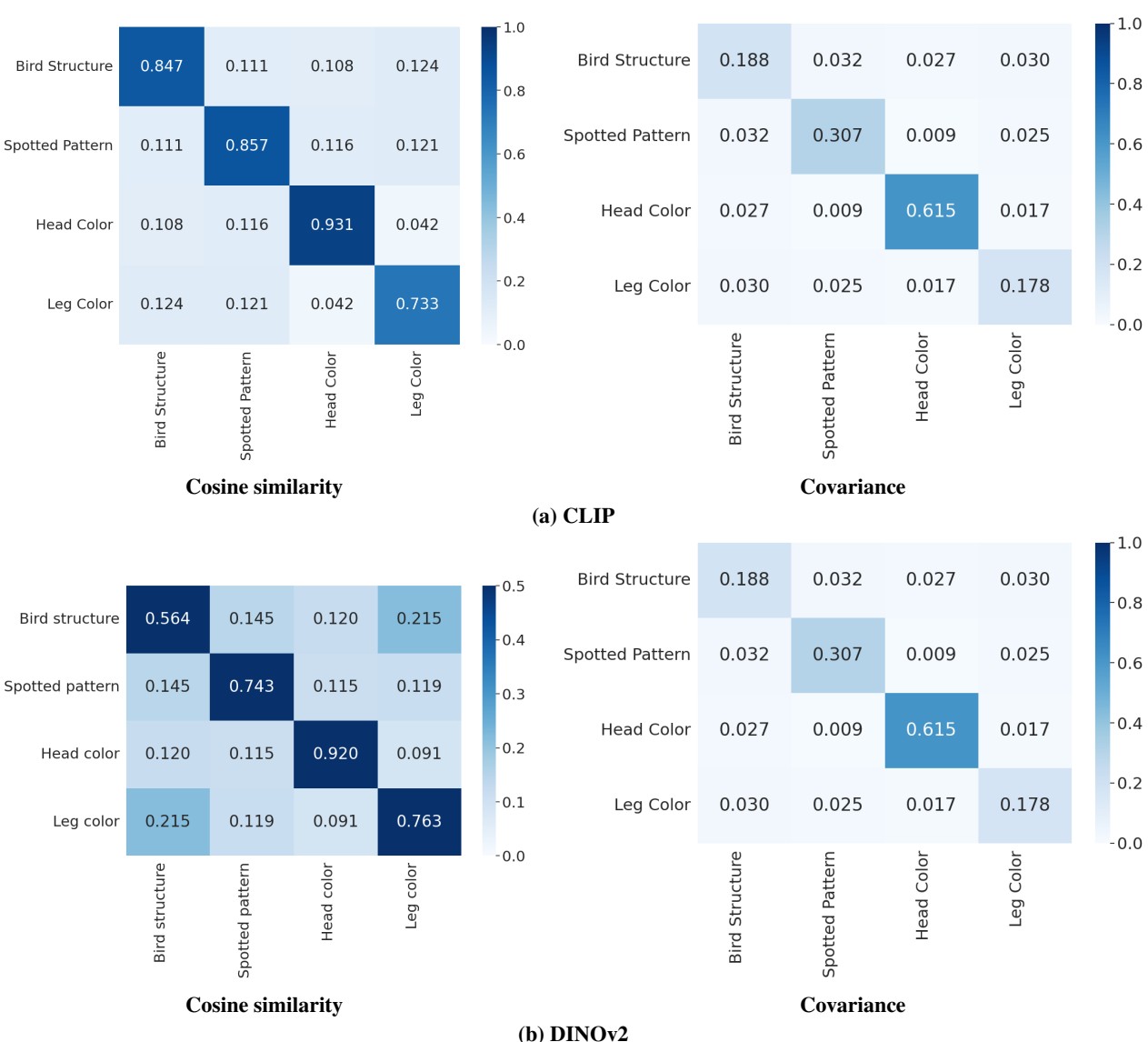

*Figure 9.* **Multitask geometric decomposition in pretrained foundation models.** For CLIP and DINOv2 representations on CUB200, we analyze the interaction between decision axes induced by different semantic tasks. **Left**: pairwise cosine similarity between task-specific decision axes, showing that distinct tasks are represented with low directional overlap. **Right**: covariance structure in the learned representation space, illustrating that substantial variance persists in directions orthogonal to the discriminative axes.

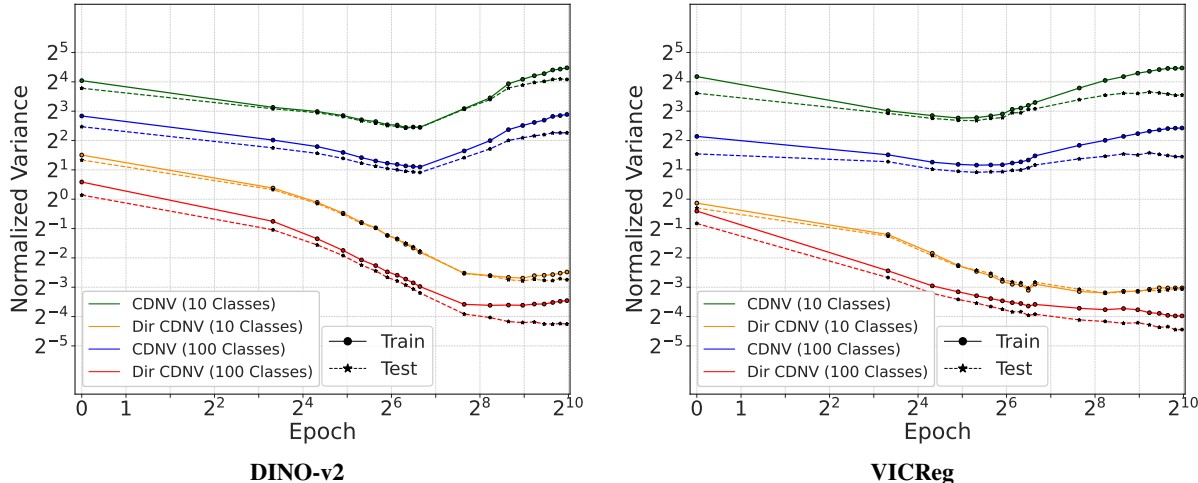

*Figure 10.* **Directional collapse manifests across semantic hierarchies.** We compare CDNV and directional CDNV at two levels of granularity: 10 semantic superclasses and 100 fine-grained classes, on both training and test data. Directional CDNV decreases much more than CDNV at both granularities, demonstrating that SSL models primarily tighten class geometry along separating directions.

$d_{ij} := \|\mu_j - \mu_i\|_2 > 0$ and define $u_{ij} := \frac{\mu_j - \mu_i}{d_{ij}}$.

**Proposition C.1** (Pairwise NCC error with tunable coefficients). *Assume $M_{4,c} := \mathbb{E}\|z_c - \mu_c\|^4 < \infty$ for $c \in \{i, j\}$, and define*

$$v_c := \text{tr}(\Sigma_c), \qquad \tilde{V}_{ij} := \frac{u_{ij}^\top \Sigma_i u_{ij}}{d_{ij}^2}, \qquad V_{ij} := \frac{v_i + v_j}{d_{ij}^2}, \qquad \Theta_{ij} := \frac{M_{4,i} + M_{4,j}}{d_{ij}^4}.$$

*Let $\Delta := \|z_i - \widehat{\mu}_j\|_2^2 - \|z_i - \widehat{\mu}_i\|_2^2$ and suppose $\mathbb{E}[\Delta] = d_{ij}^2 + \frac{v_j - v_i}{m} > 0$. For any weights $\lambda_T, \lambda_S, \lambda_Q > 0$, write $\kappa := \lambda_T + \lambda_S + \lambda_Q$ and*

$$a_T := \frac{4\kappa}{m \, \lambda_T}, \qquad a_S := \frac{\kappa}{m \, \lambda_S}, \qquad a_Q := \frac{\kappa}{m^3 \, \lambda_Q}.$$

*Then*

$$\Pr(\text{NCC predicts } j \mid i) = \Pr(\Delta \le 0) \le \frac{4\tilde{V}_{ij} + a_T V_{ij}^2 + \left(\frac{a_T}{4} + a_S\right) V_{ij} + a_Q \left(\Theta_{ij} + 2(m-1) V_{ij}^2\right)}{\left(1 + \frac{v_j - v_i}{m \, d_{ij}^2}\right)^2}. \qquad (2)$$

*Proof.* Set

$$A := z_i - \frac{\mu_i + \mu_j}{2}, \qquad X := (z_i - \mu_i)^\top u_{ij}, \qquad T := A^\top(\delta_j - \delta_i),$$
$$S := (\mu_j - \mu_i)^\top(\delta_j + \delta_i), \qquad Q := \|\delta_j\|_2^2 - \|\delta_i\|_2^2.$$

A direct expansion gives

$$\Delta = d_{ij}^2 - 2d_{ij}X - 2T + S + Q,$$

hence $\mathbb{E}[\Delta] = d_{ij}^2 + \frac{v_j - v_i}{m}$.

**Orthogonality.** We claim that $\text{Cov}(X, T) = \text{Cov}(X, S) = \text{Cov}(X, Q) = 0$. Indeed, $(\delta_i, \delta_j)$ is independent of $z_i$ and has mean 0, so

$$\mathbb{E}[XT] = \mathbb{E}\left[X A^\top(\delta_j - \delta_i)\right] = \mathbb{E}[X A^\top] \, \mathbb{E}[\delta_j - \delta_i] = 0,$$

and since $\mathbb{E}[T] = 0$, this implies $\text{Cov}(X, T) = 0$. Similarly, using independence of $X$ and $(\delta_i, \delta_j)$,

$$\mathbb{E}[XS] = \mathbb{E}[X] \, \mathbb{E}\left[(\mu_j - \mu_i)^\top(\delta_j + \delta_i)\right] = 0,$$

and since $\mathbb{E}[S] = 0$, we get $\text{Cov}(X, S) = 0$. Finally, $X$ is a function of $z_i$ while $Q = \|\delta_j\|_2^2 - \|\delta_i\|_2^2$ is a function of $(\delta_i, \delta_j)$, and $(\delta_i, \delta_j)$ is independent of $z_i$, so $X$ is independent of $Q$ and thus $\text{Cov}(X, Q) = 0$.

Let $U_1 := -2d_{ij}X$, $W_1 := -2T$, $W_2 := S$, $W_3 := Q$ and $V := W_1 + W_2 + W_3$. By bilinearity of covariance and the above orthogonality relations, $\mathrm{Cov}(U_1, V) = 0$, hence

$$\mathrm{Var}(\Delta) = \mathrm{Var}(U_1) + \mathrm{Var}(V), \qquad \mathrm{Var}(U_1) = 4d_{ij}^2 \mathrm{Var}(X) = 4d_{ij}^2 \cdot u_{ij}^\top \Sigma_i u_{ij} = 4d_{ij}^4 \tilde{V}_{ij}.$$

**Weighted variance control for $V$.** For any $\lambda_T, \lambda_S, \lambda_Q > 0$ and $\kappa := \lambda_T + \lambda_S + \lambda_Q$, the weighted Cauchy–Schwarz inequality yields

$$\mathrm{Var}(V) = \mathrm{Var}(W_1 + W_2 + W_3) \leq \kappa \left( \frac{\mathrm{Var}(W_1)}{\lambda_T} + \frac{\mathrm{Var}(W_2)}{\lambda_S} + \frac{\mathrm{Var}(W_3)}{\lambda_Q} \right).$$

Since $\mathrm{Var}(W_1) = 4\mathrm{Var}(T)$, $\mathrm{Var}(W_2) = \mathrm{Var}(S)$, $\mathrm{Var}(W_3) = \mathrm{Var}(Q)$, we have

$$\mathrm{Var}(V) \leq \kappa \left( \frac{4\,\mathrm{Var}(T)}{\lambda_T} + \frac{\mathrm{Var}(S)}{\lambda_S} + \frac{\mathrm{Var}(Q)}{\lambda_Q} \right).$$

**Component variances.** Standard moment calculus and PSD bounds give

$$\mathrm{Var}(T) = \frac{1}{m} \mathrm{tr}\left( (\Sigma_i + \Sigma_j)(\Sigma_i + \tfrac{1}{4}(\mu_j - \mu_i)(\mu_j - \mu_i)^\top) \right), \quad \mathrm{Var}(S) = \frac{1}{m}(\mu_j - \mu_i)^\top(\Sigma_i + \Sigma_j)(\mu_j - \mu_i),$$

$$\mathrm{Var}(Q) = \sum_{c \in \{i,j\}} \left[ \frac{1}{m^3}\left(M_{4,c} - v_c^2\right) + \frac{2(m-1)}{m^3} \mathrm{tr}(\Sigma_c^2) \right].$$

Using $\mathrm{tr}(AB) \leq \sqrt{\mathrm{tr}(A^2)\,\mathrm{tr}(B^2)}$, $\mathrm{tr}(\Sigma_c^2) \leq v_c^2$, and

$$(\mu_j - \mu_i)^\top(\Sigma_i + \Sigma_j)(\mu_j - \mu_i) \leq d_{ij}^2(v_i + v_j),$$

we obtain

$$\mathrm{Var}(T) \leq \frac{1}{m}\left( v_i^2 + v_i v_j + \tfrac{1}{4}d_{ij}^2(v_i + v_j) \right), \quad \mathrm{Var}(S) \leq \frac{1}{m}d_{ij}^2(v_i + v_j), \quad \mathrm{Var}(Q) \leq \frac{d_{ij}^4}{m^3}\left( \Theta_{ij} + 2(m-1)V_{ij}^2 \right).$$

Also $v_i^2 + v_i v_j \leq (v_i + v_j)^2 = d_{ij}^4 V_{ij}^2$ and $d_{ij}^2(v_i + v_j) = d_{ij}^4 V_{ij}$.

**Assemble.** Combining the pieces,

$$\mathrm{Var}(\Delta) \leq d_{ij}^4 \left[ 4\tilde{V}_{ij} + \frac{4\kappa}{m\,\lambda_T}V_{ij}^2 + \left( \frac{\kappa}{m\,\lambda_T} + \frac{\kappa}{m\,\lambda_S} \right)V_{ij} + \frac{\kappa}{m^3\,\lambda_Q}\left( \Theta_{ij} + 2(m-1)V_{ij}^2 \right) \right].$$

Finally, Chebyshev's inequality gives

$$\Pr(\Delta \leq 0) \leq \frac{\mathrm{Var}(\Delta)}{(\mathbb{E}\Delta)^2} = \frac{\mathrm{Var}(\Delta)}{\left( d_{ij}^2 + \frac{v_j - v_i}{m} \right)^2},$$

and abbreviating $a_T, a_S, a_Q$ yields (2). $\qquad\square$

**Theorem C.2.** *Let $C' \geq 2$ and $m \geq 1$ be integers. Fix a feature map $f : \mathcal{X} \to \mathbb{R}^d$ and class-conditional distributions $D_1, \ldots, D_{C'}$ over $\mathcal{X}$. We have:*

$$\mathrm{err}_{m,\mathcal{C}}^{\mathrm{NCC}}(f) \leq \frac{1}{C'} \sum_{i=1}^{C'} \sum_{j \neq i} \frac{4\tilde{V}_{ij} + \frac{12}{m}V_{ij}^2 + \frac{6}{m}V_{ij}}{\left( 1 + \frac{v_j - v_i}{m\,d_{ij}^2} \right)^2}$$

$$+ \frac{1}{C'} \sum_{i=1}^{C'} \sum_{j \neq i} \frac{\frac{3}{m^3}\left( \Theta_{ij} + 2(m-1)V_{ij}^2 \right)}{\left( 1 + \frac{v_j - v_i}{m\,d_{ij}^2} \right)^2}.$$

*Proof of Thm. C.2.* Fix $i \neq j$ and apply Prop. C.1 with $\lambda_T = \lambda_S = \lambda_Q = 1$, so $\kappa = 3$ and thus

$$a_T = \frac{4\kappa}{m\lambda_T} = \frac{12}{m}, \qquad a_S = \frac{\kappa}{m\lambda_S} = \frac{3}{m}, \qquad a_Q = \frac{\kappa}{m^3\lambda_Q} = \frac{3}{m^3}.$$

This gives the stated pairwise bound with numerator

$$4\tilde{V}_{ij} + \frac{12}{m}V_{ij}^2 + \left(\frac{3}{m} + \frac{3}{m}\right)V_{ij} + \frac{3}{m^3}\left(\Theta_{ij} + 2(m-1)V_{ij}^2\right).$$

For multiclass NCC error, use the union bound $\Pr(\hat{y}^{\mathrm{NCC}}(z_i) \neq i) \leq \sum_{j\neq i} \Pr(\Delta_{i\to j} \leq 0)$, then average over $i$ to obtain the theorem. $\qquad\square$

**Theorem 4.1.** *Let $C' \geq 2$ and $m \geq 10$ be integers. Fix a feature map $f : \mathcal{X} \to \mathbb{R}^d$ and class-conditional distributions $D_1, \ldots, D_{C'}$ over $\mathcal{X}$. Define $E_{ij}^1 := \frac{4}{m}(V_{ij}^2 + \frac{1}{4}V_{ij})$, $E_{ij}^2 := \frac{V_{ij}}{m}$, $E_{ij}^3 := \frac{\Theta_{ij}+2(m-1)V_{ij}^2}{m^3}$. Then the average multiclass error of the NCC classifier satisfies*

$$\mathrm{err}_{m,\mathcal{C}}^{\mathrm{NCC}}(f) \leq \frac{1}{C'}\sum_{i=1}^{C'}\sum_{j\neq i}\frac{4\,\tilde{V}_{ij}}{\left(1+\frac{v_j-v_i}{m\,d_{ij}^2}\right)^2}$$

$$+ \frac{1}{C'}\sum_{i=1}^{C'}\sum_{j\neq i}\frac{\left(\sqrt{E_{ij}^1}+\sqrt{E_{ij}^2}+\sqrt{E_{ij}^3}\right)^2}{\left(1+\frac{v_j-v_i}{m\,d_{ij}^2}\right)^2}.$$

*Proof of Thm. 4.1.* Fix $i \neq j$ and apply Prop. C.1. Group the $\lambda_T$-controlled terms as

$$a_T V_{ij}^2 + \frac{a_T}{4}V_{ij} = \kappa \cdot \frac{4}{m\lambda_T}\left(V_{ij}^2 + \frac{1}{4}V_{ij}\right) = \kappa \cdot \frac{E_{ij}^1}{\lambda_T},$$

and similarly

$$a_S V_{ij} = \kappa \cdot \frac{1}{m\lambda_S}V_{ij} = \kappa \cdot \frac{E_{ij}^2}{\lambda_S}, \qquad a_Q(\Theta_{ij} + 2(m-1)V_{ij}^2) = \kappa \cdot \frac{E_{ij}^3}{\lambda_Q},$$

where

$$E_{ij}^1 := \frac{4}{m}\left(V_{ij}^2 + \frac{1}{4}V_{ij}\right), \quad E_{ij}^2 := \frac{1}{m}V_{ij}, \quad E_{ij}^3 := \frac{1}{m^3}\left(\Theta_{ij} + 2(m-1)V_{ij}^2\right).$$

Hence Prop. C.1 yields

$$\Pr(\Delta_{i\to j} \leq 0) \leq \frac{4\tilde{V}_{ij} + \kappa\left(\frac{E_{ij}^1}{\lambda_T} + \frac{E_{ij}^2}{\lambda_S} + \frac{E_{ij}^3}{\lambda_Q}\right)}{\left(1 + \frac{v_j-v_i}{m\,d_{ij}^2}\right)^2}.$$

By Cauchy–Schwarz in $\mathbb{R}^3$ (take $a_k = \sqrt{\lambda_k}$ and $b_k = \sqrt{E_{ij}^k/\lambda_k}$),

$$\left(\lambda_T + \lambda_S + \lambda_Q\right)\left(\frac{E_{ij}^1}{\lambda_T} + \frac{E_{ij}^2}{\lambda_S} + \frac{E_{ij}^3}{\lambda_Q}\right) \geq \left(\sqrt{E_{ij}^1} + \sqrt{E_{ij}^2} + \sqrt{E_{ij}^3}\right)^2,$$

with equality for $\lambda_T : \lambda_S : \lambda_Q = \sqrt{E_{ij}^1} : \sqrt{E_{ij}^2} : \sqrt{E_{ij}^3}$. Choosing $\lambda_T = \sqrt{E_{ij}^1}$, $\lambda_S = \sqrt{E_{ij}^2}$, $\lambda_Q = \sqrt{E_{ij}^3}$ gives

$$\Pr(\Delta_{i\to j} \leq 0) \leq \frac{4\tilde{V}_{ij} + \left(\sqrt{E_{ij}^1} + \sqrt{E_{ij}^2} + \sqrt{E_{ij}^3}\right)^2}{\left(1 + \frac{v_j-v_i}{m\,d_{ij}^2}\right)^2}.$$

Finish by the same union bound over $j \neq i$ and averaging over $i$. $\qquad\square$

*

*Proof.* For $s, t \in \{\pm 1\}$ define the joint conditional means

$$\mu_{s,t} := \mathbb{E}[z \mid y^{(1)} = s, \, y^{(2)} = t],$$

and define the within-$y^{(1)}$ difference across $y^{(2)}$ as

$$\Gamma_s := \mu_{s,+} - \mu_{s,-}.$$

Fix any $s \in \{\pm 1\}$. Consider the scalar random variable $u_1^\top z$ conditioned on $y^{(1)} = s$. By the law of total variance,

$$
\begin{aligned}
u_1^\top \Sigma_s^{(1)} u_1 &= \mathrm{Var}(u_1^\top z \mid y^{(1)} = s) \\
&= \mathbb{E}\left[ \mathrm{Var}(u_1^\top z \mid y^{(1)} = s, \, y^{(2)}) \,\Big|\, y^{(1)} = s \right] \; + \; \mathrm{Var}\left( \mathbb{E}[u_1^\top z \mid y^{(1)} = s, \, y^{(2)}] \,\Big|\, y^{(1)} = s \right) \\
&\geq \mathrm{Var}\left( u_1^\top \mu_{s, y^{(2)}} \,\Big|\, y^{(1)} = s \right).
\end{aligned}
$$

Since the tasks are independent and $y^{(2)}$ is balanced, we have $\Pr(y^{(2)} = +1 \mid y^{(1)} = s) = \Pr(y^{(2)} = -1 \mid y^{(1)} = s) = \frac{1}{2}$. Therefore,

$$\mathrm{Var}\left( u_1^\top \mu_{s, y^{(2)}} \,\Big|\, y^{(1)} = s \right) = \frac{1}{4}\left( u_1^\top (\mu_{s,+} - \mu_{s,-}) \right)^2 = \frac{1}{4}\left( u_1^\top \Gamma_s \right)^2.$$

Combining the last two displays gives

$$|u_1^\top \Gamma_s| \leq 2\sqrt{u_1^\top \Sigma_s^{(1)} u_1} = 2d_1 \sqrt{\tilde{V}_s^{(1)}} \leq 2d_1 \sqrt{\tilde{V}^{(1)}}.$$

Next we express the second task's mean gap in terms of the joint means. Independence implies $\Pr(y^{(1)} = s \mid y^{(2)} = t) = \Pr(y^{(1)} = s) = \frac{1}{2}$, hence for each $t \in \{\pm 1\}$,

$$\mu_t^{(2)} = \mathbb{E}[z \mid y^{(2)} = t] = \sum_{s \in \{\pm 1\}} \Pr(y^{(1)} = s \mid y^{(2)} = t) \, \mu_{s,t} = \tfrac{1}{2}(\mu_{+,t} + \mu_{-,t}).$$

Thus

$$\mu_+^{(2)} - \mu_-^{(2)} = \tfrac{1}{2}\left( (\mu_{+,+} - \mu_{+,-}) + (\mu_{-,+} - \mu_{-,-}) \right) = \tfrac{1}{2}(\Gamma_+ + \Gamma_-).$$

Taking inner products with $u_1$ yields

$$d_2 \, u_1^\top u_2 = u_1^\top (\mu_+^{(2)} - \mu_-^{(2)}) = \tfrac{1}{2}\left( u_1^\top \Gamma_+ + u_1^\top \Gamma_- \right),$$

hence by the triangle inequality and the bound above (applied to $s = +$ and $s = -$),

$$|d_2 \, u_1^\top u_2| \leq \tfrac{1}{2}\left( |u_1^\top \Gamma_+| + |u_1^\top \Gamma_-| \right) \leq \tfrac{1}{2}\left( 2d_1 \sqrt{\tilde{V}^{(1)}} + 2d_1 \sqrt{\tilde{V}^{(1)}} \right) = 2d_1 \sqrt{\tilde{V}^{(1)}}.$$

Dividing by $d_2$ gives $|u_1^\top u_2| \leq 2(d_1/d_2)\sqrt{\tilde{V}^{(1)}}$. Swapping the roles of the two tasks yields $|u_1^\top u_2| \leq 2(d_2/d_1)\sqrt{\tilde{V}^{(2)}}$. Taking the minimum of the two bounds proves (**??**). □

## D. Optimality of the leading constant $4$

This section shows that the leading coefficient $4$ multiplying $\tilde{V}_{ij}$ in Theorems C.2 and 4.1 cannot be improved under only second-moment information. The key point is that, in the known-centroid limit, pairwise NCC error is a one-sided tail probability of a mean-zero scalar random variable with known variance. The sharp distribution-free bound is given by Cantelli's (one-sided Chebyshev) inequality, and it is tight via a two-point construction. Therefore any bound that depends only on $\mathbb{E}[X] = 0$ and $\mathrm{Var}(X)$ must incur the factor $4$ in the small-$\tilde{V}_{ij}$ regime, unless additional tail assumptions are imposed.

### D.1. Reduction to a one-dimensional tail event

Consider the idealized regime where the class centroids are known, equivalently $m \to \infty$ so that $\widehat{\mu}_i = \mu_i$ and $\widehat{\mu}_j = \mu_j$. For a test point $z_i \sim D_i$, define the axis projection

$$X := (z_i - \mu_i)^\top u_{ij}, \qquad \mathrm{Var}(X) = u_{ij}^\top \Sigma_i u_{ij} = d_{ij}^2 \tilde{V}_{ij}.$$

The pairwise NCC margin becomes

$$\Delta_\infty := \|z_i - \mu_j\|_2^2 - \|z_i - \mu_i\|_2^2 = d_{ij}^2 - 2d_{ij}X,$$

so the pairwise error is exactly

$$p_{i \to j}^{\mathrm{NCC}} = \mathrm{Pr}(\Delta_\infty \le 0) = \mathrm{Pr}(X \ge d_{ij}/2). \tag{3}$$

Thus, bounding $p_{i \to j}^{\mathrm{NCC}}$ reduces to bounding a one-sided tail probability for a mean-zero scalar random variable with a known variance.

### D.2. Cantelli's inequality and the sharp bound

**Lemma D.1** (Cantelli / one-sided Chebyshev). *Let $X$ be a real random variable with $\mathbb{E}[X] = 0$ and $\mathrm{Var}(X) = \sigma^2 < \infty$. Then for any $t > 0$,*

$$\mathrm{Pr}(X \ge t) \le \frac{\sigma^2}{\sigma^2 + t^2}.$$

*Moreover, this bound is tight: for any $\sigma^2 > 0$ and $t > 0$, there exists a two-point distribution on $\{-a, t\}$ (with a suitable $a > 0$) that attains equality.*

Applying Lem. D.1 to (3) with $t = d_{ij}/2$ and $\sigma^2 = d_{ij}^2 \tilde{V}_{ij}$ yields

$$p_{i \to j}^{\mathrm{NCC}} \le \frac{d_{ij}^2 \tilde{V}_{ij}}{d_{ij}^2 \tilde{V}_{ij} + d_{ij}^2/4} = \frac{4\tilde{V}_{ij}}{1 + 4\tilde{V}_{ij}} \le 4\tilde{V}_{ij}. \tag{4}$$

The last inequality is the linearization useful for small $\tilde{V}_{ij}$. Importantly, the fraction $\frac{4\tilde{V}_{ij}}{1+4\tilde{V}_{ij}}$ is the best possible distribution-free upper bound given only $\mathbb{E}[X] = 0$ and $\mathrm{Var}(X) = d_{ij}^2 \tilde{V}_{ij}$.

### D.3. Minimax tightness and the necessity of the factor $4$

**Proposition D.2** (Sharpness of the coefficient 4 under second moments). *Fix $d_{ij} > 0$ and $\tilde{V}_{ij} > 0$. Among all real random variables $X$ satisfying $\mathbb{E}[X] = 0$ and $\mathrm{Var}(X) = d_{ij}^2 \tilde{V}_{ij}$, the maximal value of $\mathrm{Pr}(X \ge d_{ij}/2)$ equals $\frac{4\tilde{V}_{ij}}{1+4\tilde{V}_{ij}}$.*

*Proof.* The upper bound is exactly Cantelli's inequality in Lem. D.1 applied with $t = d_{ij}/2$ and $\sigma^2 = d_{ij}^2 \tilde{V}_{ij}$, giving $\mathrm{Pr}(X \ge d_{ij}/2) \le \frac{4\tilde{V}_{ij}}{1+4\tilde{V}_{ij}}$. Tightness follows from the extremal two-point construction in Lem. D.1: let $X$ take values $t = d_{ij}/2$ and $-a$ with probabilities $p$ and $1 - p$, where $a = \sigma^2/t$ and $p = \sigma^2/(\sigma^2 + t^2)$ with $\sigma^2 = d_{ij}^2 \tilde{V}_{ij}$. This choice satisfies $\mathbb{E}[X] = 0$ and $\mathrm{Var}(X) = \sigma^2$, and yields $\mathrm{Pr}(X \ge t) = p = \sigma^2/(\sigma^2 + t^2) = \frac{4\tilde{V}_{ij}}{1+4\tilde{V}_{ij}}$. $\square$

As $\tilde{V}_{ij} \downarrow 0$, the worst-case probability behaves as $\frac{4\tilde{V}_{ij}}{1+4\tilde{V}_{ij}} = 4\tilde{V}_{ij} + o(\tilde{V}_{ij})$, so no uniform inequality of the form $\mathrm{Pr}(X \ge d_{ij}/2) \le c\tilde{V}_{ij}$ can hold for all distributions with $c < 4$.

### D.4. Connection to the finite-shot denominator $\left(1 + \frac{v_j - v_i}{m\, d_{ij}^2}\right)^2$

In the finite-shot setting, $\mathbb{E}[\Delta] = d_{ij}^2 + \frac{v_j - v_i}{m} = d_{ij}^2(1 + \alpha_{ij})$ where $\alpha_{ij} := \frac{v_j - v_i}{m d_{ij}^2}$, and our bounds normalize by $\left(1 + \frac{v_j - v_i}{m d_{ij}^2}\right)^2 = (1 + \alpha_{ij})^2$. Heuristically, the effective threshold in the one-dimensional reduction is rescaled from $d_{ij}/2$ to

$\frac{d_{ij}}{2}(1 + \alpha_{ij})$, so the second-moment-optimal leading coefficient scales as

$$\frac{4}{(1 + \alpha_{ij})^2} = \frac{4}{\left(1 + \frac{v_j - v_i}{m\, d_{ij}}\right)^2},$$

matching the leading term in Theorems C.2 and 4.1.

### D.5. Sharper bound under sub-Gaussian assumptions

The optimality of the leading constant $4$ holds in the distribution-free setting, where only second-moment information is assumed. Under stronger tail assumptions, however, the pairwise NCC error admits a sharper exponential bound.

In the known-centroid limit ($m \to \infty$), suppose that the projected class-conditional fluctuation along the decision axis is sub-Gaussian with variance proxy comparable to $d_{ij}^2 \widetilde{V}_{ij}$. Then the pairwise NCC error satisfies

$$p_{i \to j}^{\mathrm{NCC}} \leq \exp\left(-\frac{c}{\widetilde{V}_{ij}}\right),$$

for a universal constant $c > 0$.

Thus, under sub-Gaussian assumptions, directional CDNV controls the exponent of the pairwise error, rather than only yielding a linear upper bound. This does not contradict the optimality result of App. D, since the sub-Gaussian assumption excludes the extremal heavy-tailed constructions underlying the sharpness of Cantelli's inequality.

