# OpenReview forum: "Directional Neural Collapse Explains Few-Shot Transfer in Self-Supervised Learning"
_ICML.cc/2026/Conference — ICML 2026 regular_

### Official Review · Reviewer_FFyL · 2026-03-11

**Soundness:** 3
**Presentation:** 2
**Significance:** 3
**Originality:** 2
**Overall Recommendation:** 3
**Confidence:** 1

**Summary:**

The paper studies how learned self-supervised representations transfer well to downstream classification tasks. In particular, the authors provide a tight error bound for downstream classification when using a linear probe or the nearest class centroid classifier.

**Compliance With Llm Reviewing Policy:**

Affirmed.

**Final Justification:**

As stated in the reply to the authors’ rebuttal, although the authors provided additional clarification, the concerns have not been resolved, so I maintain the current score. But as my confidence score is low, more weight may be given to other reviewers.

**Key Questions For Authors:**

1. My understanding is that the motivation for using directional CDNV is largely inspired by prior work [a], while this paper mainly provides a sharper error bound. Could the authors clarify the additional conceptual contributions beyond what was already established in [a]? Since this is my main concern, I would be willing to increase my score if this concern arises from a misunderstanding on my part.
2. How would the analysis change if embeddings were normalized and constrained to lie on the unit sphere, which is common in many representation learning setups?

[a] Luthra, A., Yang, T., and Galanti, T. Self-supervised contrastive learning is approximately supervised contrastive learning. NeurIPS 2025.

**Limitations:**

Although the authors do not explicitly discuss the limitations or potential negative societal impacts of their work, such a discussion does not seem necessary for this paper.

**Strengths And Weaknesses:**

Strengths:
1. The paper provides sharper error bounds for downstream classification, which are well motivated by practical use cases in representation learning and few-shot transfer.
2. The paper addresses an important and widely observed phenomenon, that is why frozen representations enable strong few-shot transfer in downstream tasks.

Weaknesses:
1. Although the paper is generally well written, it may still be difficult for non-experts to follow. Adding a clearer conclusion section that summarizes the main findings and takeaways would improve readability.
2. The core idea of focusing on directional CDNV instead of classical CDNV appears to build on prior work. As a result, the novelty seems somewhat incremental, mainly in the form of a sharper error bound.
3. The paper would benefit from a clearer discussion of how the proposed metrics could be used in practice.

---

> ### Author Rebuttal · Authors · 2026-03-28
>
> We thank the reviewer for their thoughtful and constructive feedback.
>
> New experiments: https://anonymous.4open.science/r/directional-nc-rebuttal-E9F8
>
> For a summary of experiments see the first comment to reviewer TDA3.
>
> > Reviewer: Adding a conclusion section
>
> Response: We thank the reviewer for this helpful suggestion. We agree that the paper would benefit from a clearer high-level summary, especially for non-expert readers. In the revision, we will add the following conclusion section:
>
> \section{Conclusion}
>
> Our work builds on recent progress in understanding why self-supervised representations transfer well, and strengthens the case that directional collapse is central to this phenomenon. In particular, we solidify that few-shot transfer is governed not by global within-class variance, but by variance along class-separating directions.
>
> Compared to recent work (Luthra et al. 2025), we derive sharper non-asymptotic bounds for nearest-class-center classification and linear probing, in which directional CDNV appears as the leading term, together with explicit finite-shot correction terms, a separate fourth-moment contribution, and an optimal leading constant under second-moment assumptions. We also show empirically, across diverse SSL methods and architectures, that directional CDNV consistently collapses during pretraining even when classical CDNV remains large. Finally, we connect directional collapse to multitask structure by showing that small directional CDNV across independent tasks implies near-orthogonality of their decision axes, linking strong few-shot transfer to efficient packing of multiple tasks in a shared representation.
>
> Taken together, these results provide a sharper theoretical account, broader empirical support, and a new multitask perspective on why modern SSL representations support strong few-shot transfer.
>
> > Reviewer: Novelty + incremental work + relationship with [a]
>
> Response: We agree that directional CDNV itself was introduced in prior work [a], and the paper is transparent about this. However, the present paper goes beyond [a] in important ways. First, it provides a substantially sharper non-asymptotic analysis, with explicit finite-shot correction terms, a separate fourth-moment contribution, and an optimal leading constant under second-moment information. While [a] gave initial evidence that directional CDNV is relevant for few-shot transfer, its guarantee was fairly loose. Our analysis shows that directional CDNV can be the dominant term in an informative finite-shot certificate, providing stronger theoretical support for its role as a transferability metric. This requires a new derivation that accounts for higher-order terms and is not a direct extension of [a]. Second, the paper establishes a new multitask geometric consequence: simultaneously small directional CDNV across independent labelings implies near-orthogonality of the corresponding decision axes, giving a new connection between few-shot transfer and multitask compatibility. Third, the paper broadens the empirical picture from a relatively narrow SimCLR/ResNet setting to diverse SSL methods and architectures, supporting the view that decision-axis collapse is not specific to a single method or backbone, but reflects a broader way SSL models organize the embedding space. We hope this clarifies that the paper is intended not only to sharpen the theory in [a], but also to broaden and deepen the conceptual picture.
>
> > Reviewer: The paper would benefit from a clearer discussion of how the proposed metrics could be used in practice.
>
> Response: Although our main goal is to characterize the principles governing SSL transfer, the proposed metrics may also be useful in practice. Directional CDNV could serve as a tool for model selection and as a useful regularization target. More broadly, it provides a way to assess whether a representation exhibits approximate orthogonality between task-relevant directions, a structure that may support compositional generalization. This is especially relevant in domains such as robotics and embodied AI, where data is expensive and frozen representations are often used for efficient adaptation across diverse tasks [1,2].
>
> [1] R3M: A Universal Visual Representation for Robot Manipulation
>
> [2] Where are we in the search for an Artificial Visual Cortex for Embodied Intelligence?
>
> > Reviewer: How would the analysis change if embeddings were normalized in a unit sphere?
>
> Response: Our theory applies equally to normalized embeddings on the unit sphere. The main conclusion is unchanged: the relevant quantity is still variance along class-separating directions, so directional CDNV remains the key predictor of few-shot transfer. Normalization removes radial variation and makes the geometry more angular, but does not change the central distinction between total variance and decision-relevant variance. In our experiments, consistent with standard SSL practice, we evaluate normalized embeddings.

---

> > ### Author Rebuttal · Reviewer_FFyL · 2026-03-31
> >
> > Thank you for the detailed rebuttal. I appreciate the additional explanations and clarifications provided.
> >
> > However, my main concern is not fully resolved. While I understand that the paper strengthens prior analysis through sharper non-asymptotic bounds and additional technical refinements, it remains unclear to me whether these improvements translate into qualitatively new insights beyond prior work. In particular, the central conclusion that directional CDNV governs few-shot transfer appears largely consistent with earlier results, and I still find it difficult to identify a clear conceptual shift enabled by the new analysis.
> >
> > Additionally, the discussion of practical implications remains somewhat high-level. While the metric is suggested as potentially useful for model selection or regularization, it is not yet clear how it can be concretely applied, or whether it provides actionable advantages in realistic settings.

---

> > > ### Author Response · Authors · 2026-04-01
> > >
> > > Thank you for the follow-up. We agree that this distinction should be made more explicit in the paper.
> > >
> > > Our claim is not simply that we tighten a previously known bound around the same conclusion. In [a], directional CDNV was introduced as the right anisotropy-aware quantity, but the resulting guarantee was still too loose to establish that it actually governs few-shot transfer at realistic shot counts. In that sense, [a] made the directional story plausible, but not yet quantitatively decisive. What changes in our paper is that the analysis becomes sharp enough to separate three effects: the leading decision-axis term (directional CDNV), finite-shot centroid-estimation error, and higher-moment / tail effects. This matters conceptually because it tells us not only that directional CDNV is relevant, but also when it is sufficient and what explains the remaining gap. Empirically, this sharper decomposition is important because the resulting certificate is informative at practical shot sizes and tracks observed few-shot performance, whereas the earlier bound was often too loose in exactly that regime. Thus, the contribution is not merely a tighter constant, but a shift from a suggestive proxy to a quantitatively predictive explanation.
> > >
> > > There is also a second conceptual contribution that is absent from [a]. We show that when directional CDNV is simultaneously small across independent tasks, the corresponding decision axes must be nearly orthogonal. This links few-shot transfer to multitask compatibility and representation organization, and provides a geometric explanation for how one SSL representation can support multiple tasks with low interference. We view this orthogonalization result as qualitatively new because it goes beyond single-task certification and connects directional collapse to the structure of multitask reuse.
> > >
> > > The empirical scope also changes the strength of the claim. Prior evidence for the directional picture was limited to a relatively narrow SimCLR/ResNet setting. In contrast, our experiments support the same mechanism across a much broader range of SSL paradigms, architectures, datasets, and scales, and the rebuttal further strengthens this point. This allows us to make a stronger statement than [a]: not merely that directional CDNV can matter in one contrastive setup, but that suppressing variance along class-separating directions appears to be a broader mechanism by which SSL representations support few-shot transfer.
> > >
> > > On practicality, we agree that the paper does not propose a new algorithm, and we should state that more plainly. The contribution is primarily explanatory, but we do think the quantities are actionable as diagnostics. Given a frozen encoder and a small labeled downstream task, one can estimate directional CDNV together with the finite-shot correction terms to compare encoders or checkpoints, diagnose whether transfer is limited by decision-axis variability or by small-sample estimation error, and assess whether a shared representation is likely to support multiple tasks with low interference by checking whether the induced decision axes are close to orthogonal. We see this as the right level of practical implication for a theory paper: not a finished training recipe, but a sharper framework for evaluating and reasoning about representations that practitioners already use.
> > >
> > > More broadly, the paper is in the spirit of explanatory theory works such as Neural Collapse [1], whose contribution is not a new method but a conceptual framework that clarifies the structure learned by modern models and helps organize subsequent empirical and methodological work.
> > >
> > > We will revise the paper to make two points much clearer: first, that the conceptual advance is the move from a loose directional intuition to a predictive decomposition of few-shot behavior; and second, that the practical value is primarily diagnostic and comparative, especially in multitask settings.
> > >
> > > [1] Papyan et al. 2022; Prevalence of neural collapse during the terminal phase of deep learning training

---

### Official Review · Reviewer_Wxmd · 2026-03-12

**Soundness:** 3
**Presentation:** 3
**Significance:** 3
**Originality:** 3
**Overall Recommendation:** 4
**Confidence:** 2

**Summary:**

This paper introduces directional CDNV (decision-axis variance) as a key geometric quantity explaining why frozen SSL representations support strong few-shot transfer and low multitask interference. It proves sharp, non-asymptotic multiclass generalization bounds for NCC and linear probing whose leading term is exactly directional CDNV (with explicit finite-shot centroid corrections and a fourth-moment tail term).

**Compliance With Llm Reviewing Policy:**

Affirmed.

**Key Questions For Authors:**

See Weeknees

**Limitations:**

See Weeknees

**Strengths And Weaknesses:**

Strengths:

The directional viewpoint elegantly resolves the apparent tension between persistent global within-class variance in SSL and strong downstream performance; the theory cleanly isolates the decision-relevant component.

Bounds are non-vacuous at realistic m (10–500 shots), include optimal constants (Cantelli tightness shown), and separate centroid-estimation error from intrinsic variability—significant improvements over prior directional certificates.

Broad empirical coverage (SimCLR, VICReg, MAE, I-JEPA, DINO-v2, CLIP, SigLIP; ResNet & ViT backbones) and learning-dynamics tracking make the claims robust and general.

Weakness

（1）While the leading term is theoretically optimal, the finite-shot correction terms still rely on global variance quantities (Vij, V²ij). In highly anisotropic regimes typical of SSL, these corrections can dominate at small m, and the bound occasionally overestimates error by 10–20 %. A matching lower bound or sub-Gaussian refinement would strengthen the claim that directional CDNV is the dominant quantity.

（2）Prop. 4.2 assumes two independent balanced binary labelings; real downstream tasks are often correlated, imbalanced, and multi-class. The synthetic experiment (only four factors) does not fully address whether near-orthogonality survives in natural multitask settings (e.g., 20 correlated CIFAR-100 subsets).

（3）All results are confined to mini-ImageNet (or ImageNet-1K pretraining); larger-scale validation on full ImageNet, domain-shift benchmarks, or bigger models (ViT-L) is missing. Minor inconsistencies in late-training directional CDNV (slight rebound) are not analyzed or ablated.

（4）Limited discussion of why SSL implicitly minimizes directional variance (relation to augmentation-induced nuisance directions, Barlow Twins/VICReg decorrelation, etc.) and insufficient comparison with recent transferability metrics (Galanti et al. 2023b, Wang et al. 2023, Weng et al. 2025).

---

> ### Author Rebuttal · Authors · 2026-03-31
>
> We thank the reviewer for their thoughtful and constructive feedback.
>
> New experiments: https://anonymous.4open.science/r/directional-nc-rebuttal-E9F8
>
> For a summary of experiments see the first comment to reviewer TDA3.
>
> > Reviewer: Refined bound using sub-Gaussian assumptions for tighter bounds.
>
> Response: We thank the reviewer for this suggestion. Our goal was to derive a bound under minimal assumptions on the per-class embedding distributions, and without stronger assumptions the global correction terms cannot be fully removed. We agree, however, that the result can be sharpened under an additional sub-Gaussian assumption on the projected class-conditional fluctuations along the decision axis. In the known-centroid limit, if the projected margin variable has sub-Gaussian variance proxy comparable to $d_{ij}^{2}\tilde V_{ij}$, then $p^{\mathrm{NCC}}_\{i \to j\} \le \exp (-c/\tilde{V}\_{ij})$, for a universal constant $c>0$. Under such stronger tail assumptions, directional CDNV controls the exponent of the pairwise error, not just a linear upper bound. We will add this analysis to the appendix.
>
> > Reviewer: Prop. 4.2: correlated, imbalanced, and multiple classes.
>
> We agree that Prop. 4.2 is stated in a stylized setting. Its purpose is to isolate the geometric mechanism in the clearest case, not to suggest that natural multitask families are exactly independent, balanced, and binary. The same proof template extends beyond this setting. For balanced binary tasks with correlation $\rho := \mathbb{E}[y^{(1)}y^{(2)}]$, one obtains
> $$
> |u_1^\top u_2| \le \min( \frac{d_1}{d_2}\Big(|\rho| + 2\sqrt{\widetilde V^{(1)}}\Big), \frac{d_2}{d_1}\Big(|\rho| + 2\sqrt{\widetilde V^{(2)}}\Big)),
> $$
> so Prop. 4.2 corresponds to the special case $\rho=0$. Multiclass tasks can likewise be treated pairwise via one-vs-one decision axes, yielding analogous but more notationally involved statements. We will add these extensions to the appendix.
>
> Regarding independence, this is intentional: the point is not to impose independence as a simplifying assumption, but to characterize the regime in which the representation factorizes into relatively independent semantic components rather than nested or heavily overlapping ones. To make it realistic, we conducted experiments with VicReg on CelebA and CUB-200 (Exps D-E).
>
> > Reviewer: Large-scale validation, domain-shift, bigger models.
>
> Response: Following the reviews, we conducted additional experiments to validate our theory and ideas.
>
> Regarding large-scale datasets and models, see the summary of the new experiments in the first comment to reviewer TDA3.
>
> Domain shift and large-scale pretraining: Our original submission already includes domain-shift validation (Fig. 2): CLIP and SigLIP, pretrained on 400M image-text pairs and WebLI  (please see Section 5.2; L364-365), respectively, are evaluated on mini-ImageNet, showing that our theory remains predictive across train-test distribution shifts.
>
> > Reviewer: Limited discussion of why SSL implicitly minimizes directional variance.
>
> Response: Intuitively, SSL encourages invariance to augmentation-induced nuisance directions while preserving semantically stable variation; in methods such as Barlow Twins and VICReg, covariance/decorrelation regularization further suppresses redundant spread. Our main point is that, for few-shot transfer, the relevant quantity is not global variance but variance specifically along pairwise decision axes. We will expand the discussion in the paper.
>
> >  Reviewer: Comparison with recent transferability metrics (Galanti et al. 2023b, Wang et al. 2023, Weng et al. 2025).
>
> Response: Following the reviewer’s suggestion, we added Galanti et al. (2023b) to Fig. 3 (see Exp B) and expanded the discussion of Wang et al. (2023) and Weng et al. (2025). Relative to clustering-based objectives such as Weng et al. (2025), our contribution is complementary: rather than proposing a new SSL objective, we identify a geometric quantity that is highly relevant for transfer in frozen SSL representations, namely directional CDNV rather than global clustering quality. Our focus also differs from that of Wang et al. (2023): they study post-hoc transferability estimation using NC-inspired global criteria, whereas we analyze the geometry during training and derive formal generalization bounds in terms of directional variance. To empirically demonstrate why this distinction matters, we evaluated the components of Wang et al.’s metric ($S_{\texttt{vc}}$, $S_{\texttt{seli}}$, and $S_{\texttt{ncc}}$) (see Exp B; /wang23) during the training of SimCLR and DINOv2. As shown in the table, these global components behave inconsistently across epochs and models; for instance the $S_{\texttt{seli}}$ score fluctuates drastically for SimCLR while remaining at a completely different magnitude for DINOv2. We will expand the related-work discussion and include this comparison in the appendix.

---

> > ### Author Rebuttal · Reviewer_Wxmd · 2026-04-06
> >
> > My concerns have been adequately addressed

---

> > > ### Author Response · Authors · 2026-04-06
> > >
> > > We are glad to hear that!

---

### Official Review · Reviewer_TDA3 · 2026-03-12

**Soundness:** 2
**Presentation:** 2
**Significance:** 2
**Originality:** 2
**Overall Recommendation:** 4
**Confidence:** 3

**Summary:**

This paper studies the geometric structure of representations learned by self-supervised learning and proposes Directional Neural Collapse, a phenomenon where within-class variance collapses primarily along class-separating directions while remaining large in orthogonal nuisance directions. The authors introduce a key geometric quantity, directional CDNV, which measures within-class variance projected onto the decision axis normalized by class mean separation. Unlike classical CDNV, which aggregates variance across all directions, directional CDNV focuses only on variance that directly affects classification margins.

**Compliance With Llm Reviewing Policy:**

Affirmed.

**Final Justification:**

The authors have substantially addressed all three of my concerns. The experimental scope has been significantly expanded with six new experiments (Exp A-F) covering ImageNet-1K, CelebA, CUB-200, and SVHN, along with validation on pretrained CLIP and DINOv2 models, which resolves my primary concern about the evaluation being confined to mini-ImageNet. The practical significance of directional CDNV is now better supported by consistent results across these diverse settings, and the addition of LP metrics and varying-C experiments strengthens the analysis. The clarification that the theoretical assumptions apply at the downstream evaluation stage on frozen representations, rather than imposing constraints on the pretraining process, is reasonable, and the imbalanced-class experiment on SVHN provides useful additional evidence. I will adjust my score.

**Key Questions For Authors:**

See weaknesses.

**Limitations:**

yes

**Strengths And Weaknesses:**

Strengths:
The theoretical analysis is mathematically rigorous and technically sound.
The introduction clearly motivates the anisotropy problem in SSL representations, which has been widely observed. The paper is also generally well-structured.
The directional perspective on neural collapse-like phenomena in SSL appears novel, including its focus on decision-axis variance, its connection to multitask orthogonalization, and the provision of finite-shot theoretical guarantees.
Weaknesses：
The experiments are conducted almost exclusively on mini-ImageNet, with no evaluation on broader or larger-scale datasets. This limitation makes it difficult to fully demonstrate the general applicability and robustness of the proposed method.
Although the paper shows that directional CDNV decreases during training, the tightness of this measure and its practical significance are not convincingly validated.
The theoretical guarantees rely on several assumptions that may not hold in realistic semi-supervised learning scenarios, such as balanced label distributions and linear decision boundaries. These assumptions may limit the applicability of the theoretical results in more complex real-world settings.

---

> ### Author Rebuttal · Authors · 2026-03-31
>
> We thank the reviewer for their thoughtful and constructive feedback.
>
> New experiments: https://anonymous.4open.science/r/directional-nc-rebuttal-E9F8
>
> Experiments summary:
>
> Exp A: Fig. 2 for larger-scale models and datasets: MAE (ViT-L) and SimCLR (ResNet-50) trained on ImageNet-1K and VICReg ResNet-50 trained on CelebA (200k samples) and CUB200 (~12k samples).
>
> Exp B: Fig. 3 with LP and comparison with Galanti 2023 bound (/classes_2) as well as Fig. 3 with varying $C$ (/varying_C).
>
> Exp C: Fig. 4 with larger-scale models and datasets (same as in Exp A).
>
> Exp D: Fig. 5 with real datasets (CelebA and CUB200) and synthetic experiment with additional models (VICReg ResNet-50 and ViT-B). For the real data we report the value of the covariance between task labels in the legend and the bound has the new covariance term as discussed in the response to reviewer Wxmd.
>
> Exp E: We took pre-trained CLIP and DINOv2 models on CUB200. We evaluated the cosine similarity between decision axes of pairs of tasks. We also computed the covariances between labels of those paired tasks.
>
> Exp F: Experiment with imbalanced classes: trained a SimCLR ResNet-50 on SVHN (~600k images) and found that SSL still preferentially reduces variance along class-separating directions.
>
> > Reviewer: Extending the experiments beyond mini-ImageNet.
>
> Response: While many of our controlled few-shot experiments are conducted on mini-ImageNet, the empirical study is not confined to that setting. Namely, we evaluated large-scale pretrained models such as CLIP and SigLIP pretrained on 400M image-text pairs and WebLI respectively (please see Section 5.2; L364-365). Following the reviews, we have expanded the analysis by reproducing Exps A, C, and D on CelebA, Exps A, C, D and E on CUB-200, Exp F on SVHN as well as Exps A and C for models trained from scratch on ImageNet-1k. Taken together, these results show that the observed directional-collapse behavior is not tied to a single benchmark, architecture, or pretraining regime.
>
> > Reviewer: Although the paper shows that directional CDNV decreases during training, the tightness of this measure and its practical significance are not convincingly validated.
>
> Response: We agree this should be clarified. Directional CDNV is not meant to fully determine downstream error, but to capture the leading decision-relevant geometric term, with the remaining gap explained by explicit finite-shot correction and higher-moment terms in our bounds. Empirically, it is consistently more predictive of few-shot performance than classical CDNV across methods, architectures, and datasets, and we have further strengthened this evidence with additional results on CelebA, ImageNet-1K, SVHN and CUB-200 (see Exps A-D, F); and included practical large-scale encoders such as CLIP, and DINOv2 (see Exp E). This also supports its practical relevance: it can be used to compare representations, monitor pretraining, and diagnose whether improved transfer reflects reduced decision-relevant variance rather than only reduced total within-class variance.
>
> > Reviewer: The theoretical guarantees rely on several assumptions that may not hold in realistic semi-supervised learning scenarios, such as balanced label distributions and linear decision boundaries.
>
> Response: We note that our setting is not semi-supervised learning, but few-shot downstream classification using frozen self-supervised representations. Our theory applies to a fixed pretrained representation $f$ and is agnostic to how $f$ was obtained, including the pretraining procedure, architecture, and data distribution. The assumptions highlighted by the reviewer arise at the downstream evaluation stage, where we study standard settings such as balanced few-shot tasks and simple classifiers in representation space in order to obtain a clean and interpretable characterization of decision-axis variability. Our goal is therefore not to cover arbitrary downstream tasks, but to isolate the role of decision-relevant variance in this widely studied regime. We will clarify this scope more explicitly in the revision.

---

> > ### Author Rebuttal · Reviewer_TDA3 · 2026-04-04
> >
> > The authors have substantially addressed all three of my concerns. The experimental scope has been significantly expanded with six new experiments (Exp A-F) covering ImageNet-1K, CelebA, CUB-200, and SVHN, along with validation on pretrained CLIP and DINOv2 models, which resolves my primary concern about the evaluation being confined to mini-ImageNet. The practical significance of directional CDNV is now better supported by consistent results across these diverse settings, and the addition of LP metrics and varying-C experiments strengthens the analysis. The clarification that the theoretical assumptions apply at the downstream evaluation stage on frozen representations, rather than imposing constraints on the pretraining process, is reasonable, and the imbalanced-class experiment on SVHN provides useful additional evidence. I will adjust my score.

---

> > > ### Author Response · Authors · 2026-04-04
> > >
> > > Thank you so much for the thoughtful follow-up and for reassessing the paper. We also appreciate your increased score. Your feedback helped us substantially strengthen the paper. In the final version, we will incorporate the new experiments, further clarify that the theoretical assumptions concern downstream evaluation on frozen representations, and improve the presentation of the practical significance of directional CDNV.

---

### Official Review · Reviewer_mW5L · 2026-03-13

**Soundness:** 3
**Presentation:** 3
**Significance:** 3
**Originality:** 3
**Overall Recommendation:** 5
**Confidence:** 3

**Summary:**

This paper studies why frozen SSL representations transfer well to new tasks with very few labeled examples. The central argument is that a single geometric quantity, directional CDNV (within-class variance projected onto the class-separating direction), explains both strong few-shot transfer and low interference across multiple tasks. The paper proves sharp non-asymptotic generalisation bounds for NCC and linear probe classifiers whose leading term is directional CDNV, with an optimal leading constant of 4 proven via Cantelli's inequality. It further shows that small directional CDNV across two independent tasks forces their decision axes to be nearly orthogonal. Empirically, the authors demonstrate across diverse SSL methods that directional CDNV collapses during pretraining even when classical CDNV remains large, and that the derived bounds are non-vacuous at practical shot sizes where prior bounds fail.

**Compliance With Llm Reviewing Policy:**

Affirmed.

**Final Justification:**

The paper provides a sharp, non-asymptotic theoretical account of why frozen SSL representations support strong few-shot transfer, centred on directional CDNV as the dominant geometric quantity. The core contributions are technically sound and form a coherent advance over prior work: tighter bounds with explicit finite-shot corrections, a multitask orthogonality result, and broad empirical validation across SSL paradigms and architectures.

The rebuttal addressed my main concerns directly. The near-orthogonality result has been extended to realistic settings with pretrained models on real data, LP error is now monitored alongside the bound, and multi-class experiments have been added. These additions were important and I appreciate the effort.

The multitask orthogonality result feels like a genuinely new idea that others will build on, and the empirical scope is now broad enough to be convincing. The one remaining concern is clarity: several reviewers had low confidence and struggled with the paper's motivation and positioning relative to prior work. The rebuttal clarifications were helpful, but should be properly integrated into the final version.
Overall, the rebuttal moved me from a weak accept to accept.

**Key Questions For Authors:**

1. As noted in the weaknesses, the near-orthogonality result is only tested on synthetic data with one model. Could you run the same check on the real encoders already used in the paper, such as CLIP or DINOv2? This one would likely lead to an increased score if the result holds on real data.
2. All experiments use binary tasks (C = 2), which is the most favourable case for the bound. Could you report results for larger C to show the certificates remain useful in more realistic settings? This is less likely to change the score on its own, but would address a gap.
3. Do the results suggest we can improve supervised multi-task performance by encouraging variability in non-class-separating directions through auxiliary losses?

**Limitations:**

No limitations are discussed. The near-orthogonality result only applies to two-class labelings and the paper never discusses whether it extends to multi-class tasks. Given that this is presented as an explanation for general multitask transfer, the restriction is worth acknowledging.

**Strengths And Weaknesses:**

**Strengths**:
1. This paper tackles a concrete problem with prior work, and introduces non-vacuous bounds that are empirically validated.
2. The multitask orthogonality result feels like the most genuinely new idea in the paper. The claim that small directional variance across independent tasks forces their decision directions to be perpendicular is fairly intuitive, but doesn’t seem to have been formally stated before.
3. The experiments are broader than most papers in this area. Testing across contrastive, masked, distillation, and multimodal SSL methods, on both ResNet and ViT architectures, makes it much harder to dismiss the findings as artifacts of one particular setup. Previous work only showed results on SimCLR with a ResNet-50.
4. The paper connects theory, geometry, and experiments coherently, making the narrative fairly easy to follow for readers across theory and empirical ML, though some more high level explanations of the more mathematical sections would be welcome.

**Weaknesses**:
1. The near-orthogonality experiment (Figure 5) uses only SimCLR with a ResNet-18 on a controlled synthetic dataset with four visual factors. Given that this is presented as a key contribution, experiments on more realistic data or more SSL methods would strengthen this claim.
2. The multi-class bound is obtained by summing pairwise errors via a union bound, which can be quite loose for large numbers of classes. The paper doesn’t analyse how the bound degrades as C grows, and all experiments use binary tasks (C = 2), which is the most favourable setting for the bound.
3. The paper bounds LP error through NCC (since LP error is at most NCC error), but never directly validates whether the bound is useful for LP in practice. Since LP often substantially outperforms NCC, the bound may be quite loose for LP, so a direct comparison would strengthen the paper's claims.
4. Section 5.2 contains an unresolved "Fig. X" reference.
5. The paper lacks a conclusion section that summarises and contextualises the work.

---

> ### Author Rebuttal · Authors · 2026-03-31
>
> We thank the reviewer for their thoughtful and constructive feedback.
>
> New experiments: https://anonymous.4open.science/r/directional-nc-rebuttal-E9F8
>
> For a summary of experiments see the first comment to reviewer TDA3.
>
> > Reviewer: Extending the near-orthogonality experiment to SoTA models (e.g., CLIP and DINOv2) and real data
>
> Response: To evaluate our near-orthogonality claim on real-world data, we extended our analysis (Exp E) to pretrained CLIP and DINOv2 (both ViT-B) models using the CUB-200 benchmark, which provides 312 fine-grained binary attributes. We extracted features using these pretrained encoders and aggregated selected binary attributes into broader conceptual groups. When computing the cosine similarities between the decision axes of these groups, we found the off-diagonal elements to be considerably small (0.04 - 0.21), strongly supporting our claim of near-orthogonality even in practical setup. Furthermore, we calculated the ground-truth covariances between these selected groups and found a high correlation (0.869 and 0.90) between the model's cosine similarity matrix and the true covariance matrix. This demonstrates that the near-orthogonality observed in the latent space is a faithful representation of concepts that are naturally decoupled in the dataset (as predicted in our work).
>
> Additionally, we also train a VICReg with ResNet-50 backbone on CelebA and CUB-200, and validate our claims of multi-task orthogonality on selected attributes (Exp D). Similar to our synthetic experiments, we tracked the pairwise cosine similarity between selected attributes and evaluated our theoretical bound from Prop. 4.2 across training epochs. Tracking this progression illustrates how the representations naturally restructure from a random initialization into task-specific orthogonal subspaces, strongly reinforcing our theoretical arguments.
>
> > Reviewer: Prop. 4.2 in practical settings and results for larger C
>
> Response: We agree that the multiclass bound in the main text is obtained by summing pairwise errors via a union bound, and is therefore conservative when the number of competing classes is large. This choice was intentional: the goal of the theorem is to provide a general bound under minimal assumptions on the representation distribution. Without additional assumptions, one cannot in general improve the worst-case dependence on the number of competitors. Sharper multiclass refinements may be possible under stronger tail assumptions, such as joint sub-Gaussianity of the normalized competitor margins, but assessing whether such assumptions are appropriate for learned representation distributions is beyond the scope of the current paper.
>
> Since we use minimal assumptions, the bound has a limited ability to certify in the large-$C$ base. For completeness, we added an experiment (Exp B; /varying_C) where we vary $C$ and evaluate the NCC and LP as well as our bound and competitor bounds.
>
> > Reviewer: Empirical validation of LP error required for the bound
>
> Response: Following the reviews, we added LP as an additional metric we monitor in our bound estimation plots (Exp B; /classes_2).
>
> > Reviewer: Section 5.2 contains an unresolved "Fig. X" reference.
>
> Response: Thanks, we fixed it.
>
> > Reviewer: The paper lacks a conclusion section that summarises and contextualises the work.
>
> Response: We added a conclusions section. See the first response to reviewer FFyL.
>
> > Reviewer: Do the results suggest we can improve supervised multi-task performance by encouraging variability in non-class-separating directions through auxiliary losses?
>
> Response: Potentially yes, in a qualified sense. Our results suggest that strong multi-task performance benefits from reducing variability along class-separating directions while preserving variability in orthogonal directions. This is consistent with the idea that auxiliary losses could reduce interference without forcing full collapse. That said, we do not directly study such training objectives, so we view this as a future direction rather than an established claim.

---

> > ### Author Rebuttal · Reviewer_mW5L · 2026-04-04
> >
> > Thank you for the detailed responses and the substantial new experimental work. Through your responses to mine and the other reviews, my concerns have been largely addressed.
> >
> > The extension of the near-orthogonality experiment to pretrained CLIP and DINOv2 on CUB-200 is particularly welcome, with off-diagonal cosine similarities that are consistently small and a high correlation between the cosine similarity matrix and the ground-truth label covariance matrix. The VICReg experiments on CelebA and CUB-200 further reinforce this. The addition of LP as a monitored metric and the varying-C experiments address my remaining empirical concerns.
> >
> > I am raising my score to 5: Accept. One lingering concern worth flagging: several reviewers had low confidence and struggled with the motivation and positioning of the paper relative to prior work. The clarifications provided during the discussion period were very helpful but they should be properly integrated into the final version, particularly around the conceptual distinction from [a] and the practical diagnostic value of the proposed metrics, to make the paper accessible to the broader ML audience.

---

> > > ### Author Response · Authors · 2026-04-04
> > >
> > > Thank you so much for the thoughtful follow-up and for reassessing the paper. We are grateful for your feedback and for your increased score. Your comments helped us substantially strengthen the paper. In the final version, we will incorporate the new results, clarify the motivation and positioning, make the distinction from [a] (i.e., Luthra et al. 2025) more explicit, and better emphasize the practical diagnostic value of our work so that the paper is more accessible to a broader ML audience.

---

### Decision · Program_Chairs · 2026-04-30

**Decision:**

Accept (regular)

**Comment:**

This paper asks which geometric properties are foundational for strong self-supervised generalization. The paper received 4 reviews, with 3 voting for accept (4,4,5) and 1 for reject (3). All reviewers find the theoretical framework and rigor a strength of the paper. Others strengths include useful bounds and empirical coverage. The AC focused specifically on the one reviewer without an accept rating. The AC find that the reviewer has low confidence in the their assessment. Their final concerns are that the paper is too close to existing literature and does not provide clear enough practical implications. The first concern contradicts the opinions of the other reviewers and the AC finds that final rebuttal clearly outlines why the paper is new. Regarding practical implications, the authors acknowledge that this is not within the scope of the paper and the AC agrees. As such, the AC finds no fundamental reasons for rejection and agrees with the majority vote for accept.